# Myeloid-derived grancalcin instigates obesity-induced insulin resistance and metabolic inflammation in male mice

Tian Su[1,4], Yue He[1,4], Yan Huang[1], Mingsheng Ye[1], Qi Guo[1], Ye Xiao [1], Guangping Cai[1], Linyun Chen[1], Changjun Li [1], Haiyan Zhou [1] ✉ & Xianghang Luo[1,2,3] ✉

The crosstalk between the bone and adipose tissue is known to orchestrate metabolic homeostasis, but the underlying mechanisms are largely unknown. Herein, we find that GCA + (grancalcin) immune cells accumulate in the bone marrow and release a considerable amount of GCA into circulation during obesity. Genetic deletion of Gca in myeloid cells attenuates metabolic dysfunction in obese male mice, whereas injection of recombinant GCA into male mice causes adipose tissue inflammation and insulin resistance. Mechanistically, we found that GCA binds to the Prohibitin-2 (PHB2) receptor on adipocytes and activates the innate and adaptive immune response of adipocytes via the PAK1-NF-κB signaling pathway, thus provoking the infiltration of inflammatory immune cells. Moreover, we show that GCA-neutralizing antibodies improve adipose tissue inflammation and insulin sensitivity in obese male mice. Together, these observations define a mechanism whereby bone marrow factor GCA initiates adipose tissue inflammation and insulin resistance, showing that GCA could be a potential target to treat metainflammation.

Obesity is increasingly prevalent due to lifestyle and diet modification, which include a reduction in physical activity and an oversupply of high-calorie foods. A considerable amount of research has indicated that in individuals with obesity, chronic low-grade inflammation occurs in adipose tissue that can progress to systemic inflammation[1,2]. Furthermore, obesity-induced inflammation is an important pathogenic mediator of many metabolic diseases, including cardiovascular disease, T2DM, and insulin resistance[3–5]. Hence, there is a critical need to identify the key factors that affect adipose tissue inflammation.

Bone is not only a hematopoietic organ but also served as an endocrine organ that modulates the function and metabolism of the whole body. Bone marrow-derived neutrophils participated in mediating insulin resistance of diet-induced obesity (DIO) mice by

secreting elastase[6] Hematopoietic-derived Galectin-3 can inhibit insulin signaling in adipose tissue, muscle, and liver, leading to systemic insulin resistance[7]. In addition, bone-derived other factors such as osteocalcin and osteopontin serve significant roles in the regulation of systemic glucose and energy homeostasis[8,9]. These findings indicate that there exists an active bone-adipose axis that regulates metabolic function of adipose tissue, while the underlying mechanisms still await investigation.

Aging and obesity share biological hallmarks related to a chronic low-grade inflammation status[10]. Moreover, obesity accelerates the aging process[11]. In our previous study, we found that during aging proinflammatory immune cells, including neutrophils and monocytes-macrophages, secreted grancalcin (GCA) to influence the fate of

[1]Endocrinology Research Center, Xiangya Hospital of Central South University, Changsha, Hunan 410008, China. [2]National Clinical Research Center for Geriatric Disorders, Xiangya Hospital, Changsha, Hunan 410008, China. [3]Key Laboratory of Organ Injury, Aging and Regenerative Medicine of Hunan Province, Changsha, Hunan 410008, China. [4]These authors contributed equally: Tian Su, Yue He. ✉e-mail: hyzhou02@csu.edu.cn; xianghangluo@csu.edu.cn

BMSCs[12]. Therefore, we conjectured that myeloid-derived GCA may be involved in adipose tissue inflammation and insulin resistance in obesity.

In this study, we report that GCA+ immune cells accumulated in the bone marrow during obesity, which secret abundant GCA and instigates adipose inflammation and insulin resistance. We found that GCA activated innate and adaptive immune response of adipocytes via Prohibitin-2(PHB2)-PAK1-NF-κB signaling pathway, thus provoking the infiltration of inflammatory immune cells. Furthermore, we synthesized a GCA-neutralizing antibody and rendered that its treatment of obese mice improved adipose tissue inflammation and insulin sensitivity. Together, these observations define a mechanism whereby bone marrow factor GCA initiates adipose tissue inflammation and insulin resistance, showing that GCA could be a potential target to treat metainflammation.

## Results

### Obesity induces a dynamic increase of GCA⁺ immune cells in the bone marrow

In line with the previous reported tissue distribution[13,14], the GCA protein levels were highly expressed in bone marrow, lowly expressed in adipose tissue, liver and muscle, but nearly non-existent in the spleen. Moreover, the expression level of GCA was higher in bone marrow from HFD mice than NCD subjects. Compared with other tissues, bone marrow GCA rose most significantly after HFD challenge (Supplementary Fig. 1a–b).

To assess the association of GCA expression with obesity, we examined the expression levels of GCA in the bone. The bone marrow GCA mRNA and protein levels, immunofluorescent expression in DIO (high-fat diet-induced obesity) mice and ob/ob mice presented a rise compared with their corresponding controls (Fig. 1a, b and Supplementary Fig. 1c–f).

As GCA is a secreted protein, we isolated the serum of male mice exposed to either a normal chow diet (NCD), 1 week of HFD, 2 weeks of HFD, 4 weeks of HFD, 8 weeks of HFD, or 12 weeks of HFD to detected the GCA concentration in circulation. We found that GCA content increased within 2 weeks of HFD and further elevated until 12 weeks of HFD (Fig. 1c). These data indicated that the dynamic increase of GCA is an early response to HFD.

Despite the fact that GCA levels were increased during HFD feeding, it remains unclear which cells secreted GCA during this process. Therefore, we conducted a single-cell RNA sequencing (scRNA-seq) in bone marrow cells from HFD and NCD mice. We found that *Gca* was mainly expressed in neutrophils irrespective of whether mice were fed with HFD or NCD. However, the number of GCA⁺ neutrophils was higher in bone marrow from HFD mice compared with bone marrow from NCD subjects (Fig. 1d and Supplementary Fig. 1g). In addition, compared with GCA⁻ neutrophils, GCA⁺ neutrophils express higher levels of inflammation and immune-related genes (Supplementary Fig. 1h). To further characterize the GCA⁺ neutrophils, we performed single-cell transcriptional profiling to subdivide bone marrow neutrophil populations. The results exhibited 8 distinct clusters (Fig. 1e and Supplementary Fig. 1i). In addition, clusters 1, 2, and 3 contained more GCA⁺ neutrophils in HFD mice (Fig. 1e and Supplementary Fig. 1i). Thus, we further analyzed the transcriptional profiles of these 3 clusters, and found that inflammation-related genes, including *Il1b, Fgl2, Ifitm1, Pla2g7, Ptgs2, Jaml, H2-Q10, Nlrp3, Mmp8, Fpr1, Timp2, Il1f9, Hacd4, Zmpste24, Fmnl2, Lbp*, and *Orm1* were elevated in these clusters (Fig. 1f). These data implied that GCA⁺ neutrophils increased in HFD animals and presented high levels of pro-inflammation-related genes.

With regard to human studies, participants with obesity showed even higher serum GCA contents than lean participants (Fig. 1g and clinical data in Table S1). As inflammation-associated factors and HOMA-IR were also elevated in individuals with obesity[15], we analyzed the correlation between GCA and HOMA-IR, serum IL6,TNFα and MMP2, and found that GCA expression levels displayed a positive correlation with HOMA-IR, IL6 TNFa and MMP2 (Fig. 1h–k), further implicating a role of GCA with insulin resistance and metainflammation in human.

### GCA deficiency in myeloid lineage ameliorates adipose tissue inflammation and glucose metabolism

Enthralled by the potential link between GCA and obesity, we next investigated whether deleting *Gca* in the myeloid lineage by crossing *Gca*^flox/flox mice with *Lyz2*-Cre mice (hereafter referred to as *Gca-Lyz2*-CKO) affected adipose tissue metabolism. The results in Fig. 2a and Supplementary Fig. 2a validated that *Gca* was successfully deleted from myeloid lineages, and also intimated that myeloid lineage is the main cellular source of GCA. The *Gca-Lyz2*-CKO mice and their corresponding control literates were fed an HFD for 12 weeks, meanwhile the body weight was constantly monitored. Surprisingly, GCA deficiency in the myeloid lineage had minimal effect on body weight, fat mass and daily food intake (Supplementary Fig. 2b–d). However, we observed lower fasting blood glucose, fasting insulin, and HOMA-IR index in *Gca-Lyz2*-CKO mice compared with WT littermates (Fig. 2b–d). The results of the glucose tolerance test (GTT) and insulin tolerance test (ITT) also revealed improved glucose tolerance and insulin sensitivity (Fig. 2e–i). In accordance with higher insulin sensitivity, the insulin-stimulated activation of pAKT (AKT phosphorylation) in eWAT (epididimal white adipose tissue), liver and muscle of *Gca-Lyz2*-CKO mice was intensified (Fig. 2j, k). However, deletion of *Gca* in myeloid cells displayed no effects on muscle morphology as revealed by HE staining (Supplementary Fig. 2e).

Additionally, the mRNA levels of pro-inflammatory markers in eWAT, adipocyte and liver of *Gca-Lyz2*-CKO mice were declined when compared with WT mice (Fig. 2l–n). Although we observed higher expression of insulin action-related genes in adipocytes of *Gca-Lyz2*-CKO mice, the mRNA expression level of mitochondrial biogenesis, lipolysis and lipogenesis were displayed no significant difference (Supplementary Fig. 2f). Consistently, *Gca-Lyz2*-CKO mice fed with HFD also had reduced numbers of macrophages infiltration and lower ratio of M1/M2 in the eWAT as shown by immunohistochemistry and flow cytometry analyses (Fig. 2o). The gating strategies for the macrophage flow cytometry were illustrated in Supplementary Fig. 2g.

It is well-established that inflammatory T cells infiltrate into adipose tissues in the early stages of obesity and affect the recruitment and activation of other inflammatory cells into VAT (visceral adipose tissue)[16,17]. Accordingly, we observed decreased expression of the pro-inflammatory Th1 marker genes *Tbx21* and *Ifng* in Stromal Vascular Fraction (SVF) of eWAT from *Gca-Lyz2*-CKO mice fed with HFD. The expression levels of anti-inflammatory Th2 (*Gata3*) and Treg (*Foxp3*) marker genes were also increased, albeit the differences were not significant (Fig. 2p). The gating strategies for the T cells flow cytometry were illustrated in Supplementary Fig. 2h. The results of flow cytometry analyses further confirmed decreased proportion of Th1 cells and elevated proportion of Treg cells in SVF of *Gca-Lyz2*-CKO mice (Fig. 2q), which indicated reduced activation of pro-inflammatory T cell in adipose tissue. The ratio of CD8+ to CD4 + T cells displayed no significant difference between two groups (Supplementary Fig. 2i). These combined data suggest that GCA exerts an essential role in obesity-induced inflammation and insulin resistance.

### GCA exacerbates adipose tissue inflammation, insulin resistance and glucose intolerance in vivo

To explore whether GCA was sufficient to induce WAT inflammation, male C57BL/6 mice fed with a normal chow diet (NCD) were injected with recombinant GCA (rGCA, 250 nM,100 μl) once per week for 8 weeks via tail vein injection. As expected, GCA treatment did not influence body weight, liver mass and body composition, glucose tolerance and insulin tolerance in these non-obese animals fed with

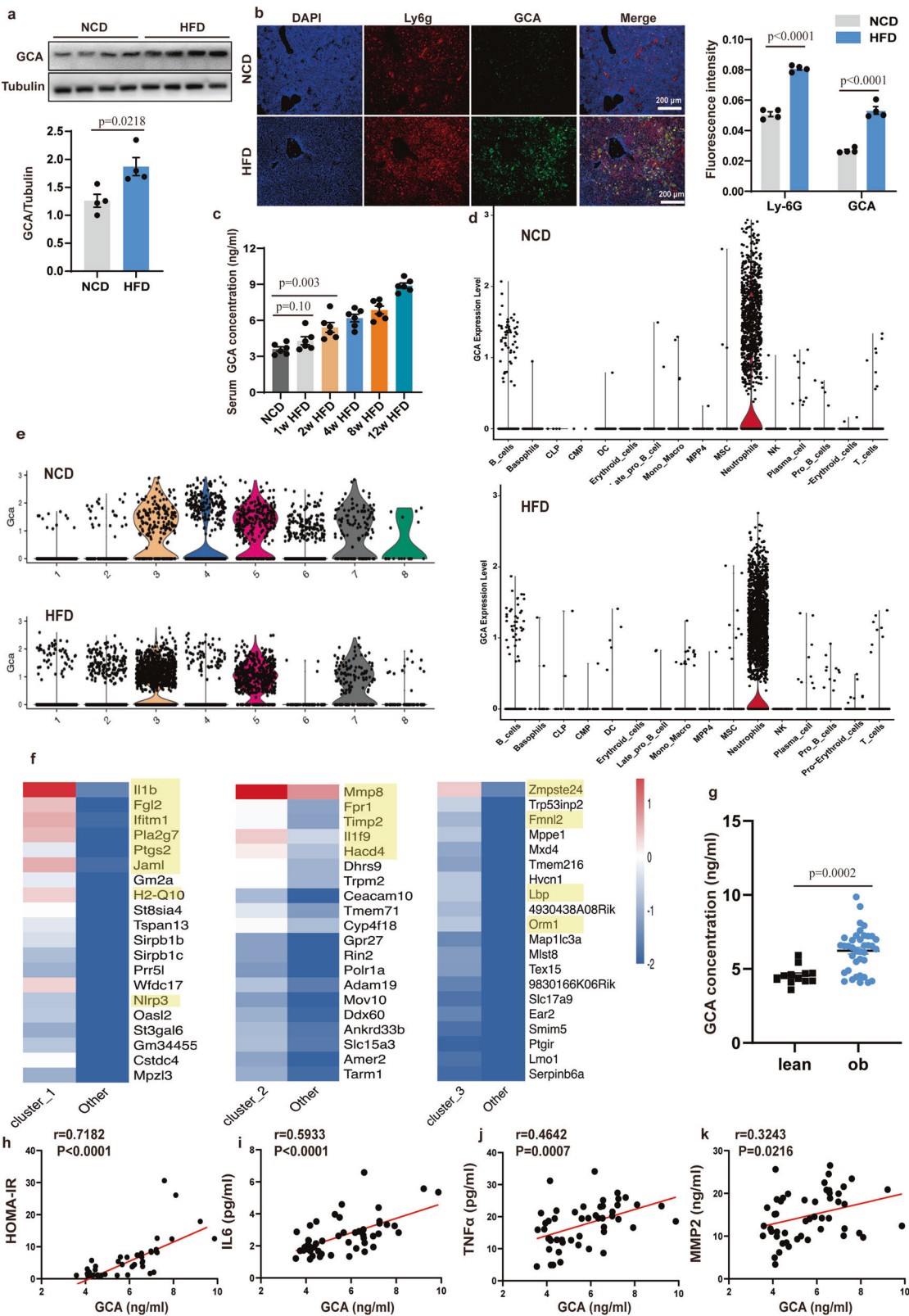

NCD (Supplementary Fig. 3a–d). Whereas the inflammatory cytokine gene expression levels were increased in eWAT and liver of GCA-treated mice (Fig. 3a and Supplementary Fig. 3e). Despite that the mRNA expression level of mitochondrial biogenesis, lipolysis, lipogenesis and insulin action-related genes in adipocytes displayed no deference between two groups, the inflammatory markers in adipocytes of GCA-treated mice were also elevated (Supplementary Fig. 3f). Furthermore, GCA-treated mice harbored more M1 macrophages in eWAT than PBS-treated mice (Fig. 3b). QPCR analysis also detected higher expression level of pro-inflammatory Th1 marker genes in SVF

**Fig. 1 | Obesity induces a dynamic increase of grancalcin + (GCA + ) immune cells in the bone marrow. a** Western Blot of grancalcin (GCA) protein level in the bone marrow of high-fat diet (HFD)-induced mice (n = 4) (top, representative pictures of Western blot; bottom, quantitative measurements of GCA proteins). **b** Immunofluorescent staining of GCA (green) and Ly6g(red) in the bone section of normal chow diet (NCD) or HFD-induced mice (n = 4) (left, representative pictures of Immunofluorescent staining; right, quantitative measurements of Ly6g and GCA proteins). The nucleuses were stained with DAPI. Scale bar, 200 μm. **c** Enzyme-linked Immunosorbent Assay (ELISA) of serum GCA concentrations in male C57BL/6 mice fed NCD or HFD for 1, 2, 4, 8, or 12 weeks (n = 6). **d** Violin plots of log-transformed gene expression of *Gca* genes in cell populations of NCD mice and HFD mice. **e** Violin plots of log-transformed gene expression of *Gca* in different clusters of bone marrow neutrophils from NCD mice and HFD mice. (**f**) The heatmap of 20 most upregulated genes in clusters 1, 2, and 3 defined in (**e**). **g** Serum GCA concentrations in individuals with (n = 38) or without (n = 12) obesity. The characteristics of the study population are provided in the Table S1. **h**–**k** Correlation analysis between GCA and homeostasis model assessment of insulin resistance (HOMA-IR), serum interleukin 6 (IL6), tumor necrosis factor α (TNFα), matrix metallopeptidase 2 (MMP2) in human participants (n = 50). Data are presented as means ± SEM. n indicates the number of biologically independent samples examined. Statistical analysis was assessed by two-sided Student's *t* test (**a, b, g**), One-way ANOVA with Tukey's multiple-comparison test (**c**) or two-sided Spearman's correlation (**h**–**k**) and significant differences were indicated with p values. Source data are provided as a Source Data File.

of mice treated with GCA (Fig. 3c). Although, the ratio of CD8+ to CD4 + T cells displayed no significantly difference between two groups, the results of flow cytometry analyses revealed that GCA treatment increased the frequency of Th1 cells (Fig. 3d and Supplementary Fig. 3g).

Then, the impact of GCA under circumstances with increased inflammation was evaluated in mice fed a high-fat diet (HFD) for 8 weeks where GCA or PBS was administrated simultaneously. Consistently, the body weight, and fat mass displayed no significant difference between two groups (Supplementary Fig. 4a, b). However, the levels of inflammatory markers in the eWAT and liver were elevated in GCA treated group (Fig. 4a and Supplementary Fig. 4c). Although, the mRNA expression levels of mitochondrial biogenesis, lipolysis and lipogenesis-related genes in adipocytes were unchanged in two groups, mice treated with GCA exhibited higher expression levels of genes involved in inflammation and insulin action than those treated mice PBS (Supplementary Fig. 4d). Meanwhile, GCA-treated mice also showed more macrophages infiltration and a higher proportion of M1 macrophages in eWAT than PBS-treated mice (Fig. 4b). In accordance with these, GCA treated mice presented a rise in pro-inflammatory Th1 cells and a reduction in anti-inflammatory Treg cells (Fig. 4c, d). However, the ratio of CD8 + T cells to CD4 + T cells was unchanged between the two groups (Supplementary Fig. 4e).

Along with the increased inflammation, GCA-treated group mice manifested elevated fasting blood glucose, fasting insulin and HOMA-IR index (Fig. 4e–g). Accordingly, administration of GCA exhibited glucose intolerance and insulin resistance as showed by GTT and ITT (Fig. 4h-k). In line with impaired insulin sensitivity, the AKT phosphorylation in the eWAT, liver and muscle upon insulin stimulation was reduced in GCA treated mice (Fig. 4l). Whereas rGCA treatment displayed no effects on muscle morphology as revealed by HE staining (Supplementary Fig. 4f). Altogether, these combined data suggest that GCA instigates adipose tissue inflammation and accelerates metabolic imbalance.

## GCA magnifies inflammation in adipocytes in vitro
A plethora of studies manifested that hypertrophic adipocyte, which induce the production of multiple adipokines and instigate pro-inflammatory Th1 cell accumulation in WAT, are key drivers of adipose tissue inflammation during obesity[18,19]. Since the content of GCA is increased in the serum two weeks after HFD feeding, a time early before macrophages infiltrate adipose tissues[20], we therefore speculated that GCA may initiate adipose inflammation by acting on adipocytes. To study the effects of GCA on adipocyte function in vitro, we incubated mouse primary preadipocytes or 3T3-L1 adipocytes with different concentrations of GCA throughout the 14-day-long differentiation process and found that GCA incubation had no effects on adipocytes differentiation (Supplementary Fig. 5a). QPCR analyses showed that the expression of *Ccl2, Il1b, Il6*, and *Tnfa* was strengthened by GCA in a concentration-dependent manner in both mouse primary adipocytes and 3T3-L1 adipocytes (Fig. 5a–d and Supplementary Fig. 5b–e). Further evaluation of the time-induced effects was

conducted using 100 nM GCA in the culture medium. The results displayed that mRNA levels of *Ccl2, Il1b, Il6, and Tnfa* were intensified by GCA treatment in a time-dependent time (Fig. 5e–h and Supplementary Fig. 5f–i). Consistently, increased protein levels of pro-inflammatory factors were also observed (Fig. 5i–l and Supplementary Fig. 5j–m).

Adipocyte exerts essential roles in the early stages of inflammation not only via the production of cytokines and chemokines but also by upregulating MHCII (class II major histocompatibility complex) to activate CD4 + T cells[21–23]. Notably, consistent with a promoting role of GCA in Th1 cell accumulation, we observed a dynamic increase of MHCII family genes in response to HFD challenge (increased at 2 weeks of HFD), parallel with the dynamic accumulation of GCA concentration (Fig. 1c). Furthermore, we also found that GCA treatment induced the expression of MHCII family genes in adipocytes in vitro (Fig. 5m). To further determine that GCA-treated adipocytes activate Th1 cells in an MHCII-restricted dependent manner, GCA-pretreated or untreated-3T3-L1 adipocytes were co-cultured with ovalbumin (OVA)-specific T cells in the presence or absence of OVA. After 48 h of co-culture, cell supernatants were collected for ELISA-based assay, and T cells were analyzed using the flow cytometry (Fig. 5n). IFNγ, which is principally produced by Th1 cells after MHCII-restricted activation, was increased in cell supernatants of GCA + OVA-treated group compared with that of GCA-treated group or OVA-treated group (Fig. 5o). Accordingly, the proportion of Th1 cells was also augmented in GCA + OVA-treated group (Fig. 5p and Supplementary Fig. 5n). Altogether, these data indicate that GCA induces the expression of inflammatory cytokines and MHCII in adipocyte, thus activating pro-inflammatory Th1 cell to magnify inflammation.

## PHB2 is a functional receptor of GCA in adipocytes
To identify the functional cell-surface receptor of GCA in adipocytes, firstly, we incubated proteins isolated from the cytomembrane of differentiated 3T3-L1 adipocytes with His-tagged GCA. The proteins pulled down by the nickel beads with His-tagged GCA were then performed mass spectrometry (Supplementary Fig. 6a). A total of 289 membrane proteins that bound to His-tagged GCA were identified. Among the top five candidates, we focused our investigations on PHB2 (Fig. 6a and Supplementary Fig. 6b), which has been reported to participate in the pathogenesis of various inflammatory disorders[24,25].

To verify the direct binding of GCA to PHB2 in vitro, Myc-tagged GCA and HA-tagged PHB2 were transfected into human embryonic kidney (HEK) 293 T cells via plasmids transfection. And then the HEK 293 T cells were collected to conduct IP(Immunoprecipitation) by anti-Myc antibody. The results of Western Blot showed a strong band of HA staining in the Myc immunoprecipitants, which indicated the interaction of GCA and PHB2(Fig. 6b). Conversely, using an anti-HA antibody to perform the IP experiment in cell lysate and anti-Myc antibody in the following WB assay also presented the binding of GCA and PHB2 (Fig. 6c).

We next investigated how PHB2 interacts with GCA. Transmembrane helix prediction algorithms predicted PHB2 to be a multiple

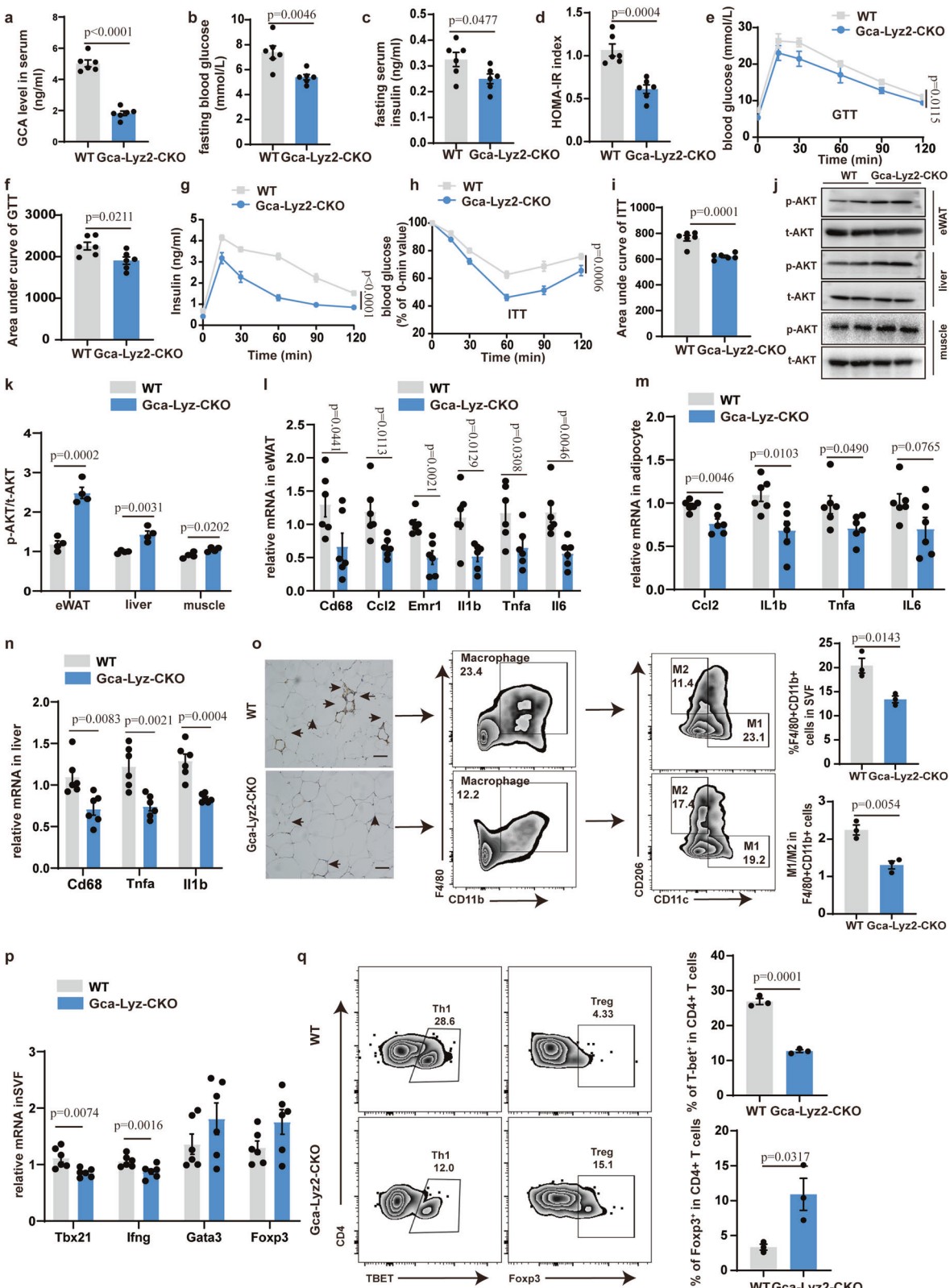

transmembrane hydrophilic protein, with multiple hydrophobic centers (hydrophobicity score greater than 0) and hydrophilic ends (Supplementary Fig. 6c). Additionally, immunofluorescence staining also confirmed that PHB2 located on the plasma membrane of differentiated 3T3-L1 adipocytes (Fig. 6d).

To further identify the exact binding domain between GCA and PHB2, we constructed a range of deletion mutants according to different domain positions. Deletion of nucleotides 1-50, 92-127, and 158-193 in GCA abolished its interaction with PHB2, indicating that domains 1-50, 92-127, and 158-193 could bind to the PHB2(Fig. 6e). Unexpectedly, deletion of either domain in PHB2 didn't affect the binding with GCA (Fig. 6f). A possible reason for this was likely that GCA binds multiple domains of PHB2 simultaneously, and deletion of a single domain has little effect on its binding.

**Fig. 2 | GCA deficiency in myeloid lineage ameliorates adipose tissue inflammation and glucose metabolism. a** Serum GCA concentrations in Gca-Lyz2-CKO and WT mice ($n = 6$). **b–d** Fasting blood glucose, fasting insulin and HOMA-IR index of Gca-Lyz2-CKO and WT mice fed with HFD for 12 weeks ($n = 6$). **e, f** Blood glucose (**e**) and area under curve (**f**) of Gca-Lyz2-CKO and wild type (WT) mice fed with HFD for 12 weeks during intraperitoneal glucose tolerance test (GTT) ($n = 6$). **g** Serum insulin concentrations during GTT ($n = 6$). **h** Blood glucose in Gca-Lyz2-CKO and WT mice fed with HFD for 12 weeks during intraperitoneal insulin tolerance test (ITT) ($n = 6$). **i** Area under curve (AUC) of Gca-Lyz2-CKO and WT mice fed with HFD for 12 weeks during ITT ($n = 6$). The values of the y-axis in AUC are the absolute measured blood glucose concentrations. **j, k** Western Blot of insulin-stimulated AKT phosphorylation in epididymal white adipose tissue (eWAT), liver and muscle **j** and quantitation of pAKT/tAKT **k** from Gca-Lyz2-CKO and WT mice fed with HFD for 12 weeks ($n = 4$). **l–n** Quantitative Polymerase Chain Reaction (QPCR) analysis of

inflammatory cytokine gene expression levels in eWAT, adipocyte and liver of Gca-Lyz2-CKO and WT mice fed with HFD for 12 weeks ($n = 6$). **o** F4/80 immunohistochemistry and flow analyses of macrophage in eWAT from Gca-Lyz2-CKO and WT mice fed with HFD for 12 weeks ($n = 3$). Scale bar, 250 μm. **p** QPCR analysis of the proinflammatory Th1 marker genes (*Tbx21* and *Ifng*), Treg (*Foxp3*) and Th2(Gata3) in SFV of Gca-Lyz2-CKO and WT mice fed with HFD for 12 weeks ($n = 6$). **q** Flow cytometry analysis of CD4+Tbet+ Th1 cells and CD4+Foxp3+ Treg cells in eWAT from Gca-Lyz2-CKO and WT mice fed with HFD for 12 weeks ($n = 3$). Data are presented as means ± SEM. n indicates the number of biologically independent samples examined. Statistical analysis was assessed by two-sided Student's *t* test (**a–d, f, i** and **k–q**) or two-way ANOVA followed with Sidak's multiple comparisons test (**e,g** and **h**) and significant differences were indicated with *p* values. Source data are provided as a Source Data File.

To prove that GCA magnifies inflammation in adipocytes via PHB2, adipocyte-specific *Phb2* knockdown mice (designated as *Phb2*-CKD mice) and control mice were generated by injecting with adeno-associated virus serotype 9, which carries *aP2* promoter-driven small hairpin RNAs targeting Phb2 (AAV9-shPhb2) or negative control (AAV9-shNC) with a Green fluorescent protein (GFP) reporter gene. Next, we confirmed that Phb2 gene expression was lower in adipocytes from *Phb2*-CKD mice compared with control mice (Supplementary Fig. 6d). Then the *Phb2*-CKD mice and control mice were fed an HFD for 8 weeks where GCA or PBS was administrated simultaneously. As expected, administration of GCA promoted the expression level of inflammatory markers in the eWAT (Fig. 6g). Moreover, mice treated with GCA also had increased macrophages infiltration and M1 polarization (Fig. 6h). Whereas silencing *Phb2* abolished the proinflammation function of GCA (Fig. 6g, h). In addition, the knockdown of *Phb2* also alleviated glucose and insulin intolerance in GCA-treated mice (Fig. 6i-o). Altogether, these results suggest that PHB2 is a functional receptor of GCA in adipocytes.

### GCA promotes phosphorylation of PAK1- NF-κB downstream of PHB2 signaling

We are still interested in the possible effect of GCA on the PHB2-signaling pathway, therefore we conducted global, quantitative phosphoproteomic analysis in differentiated 3T3-L1 adipocytes cells treated with GCA or siRNA-*Phb2* (Fig. 7a). Data in GCA versus NC (negative control) displayed 633 up-regulated phosphorylation sites on 506 proteins and 646 down-regulated phosphorylation sites on 462 proteins. However, the knockdown of *Phb2* by siRNA resulted in 472 up-regulated phosphorylation sites on 386 proteins and 182 down-regulated phosphorylation sites on 148 proteins compared with siRNA-NC treated groups (Fig. 7b, c). As kinases play a wide range of roles in cellular signaling transduction, we focus on kinases involved in the pathogenesis of various inflammatory disorders. Among these kinases, phosphorylation of PAK1 was reported to influence systemic inflammatory response in various diseases[26,27]. Activating PAK1 leads to the activation of NF-κB inflammatory signaling[28,29]. Remarkably, we observed an activation of the PAK1 and increased downstream phosphorylation of NF-κB1 (Nuclear factor NF-kappa-B p105 subunit), transcription factor AP-1 (JUN) in GCA treated group(Fig. 7d). To detect whether GCA could promote the formation of a PAK1-PHB2 complex, we conducted IP and found that PHB2 could bind to PAK1, and GCA treatment strengthened the binding of PHB2 and PAK1(Fig. 7e). The results of western blot further confirmed that GCA treatment elevated the phosphorylation levels of PAK1 and its downstream target NF-κB p65, which were abolished in siRNA-*Phb2*-transfected 3T3-L1 cells (Fig. 7f).

We also observed increased nuclear translocation of p65 in GCA treatment group as the immunofluorescence assay showed. On the contrary, inhibition of PAK1 activity by small molecule inhibitors IPA-3 blocked the nuclear translocation of p65 induced by GCA (Fig. 7g).

Coincident with these, inhibition of PAK1 also weakened the phosphorylation of p65 induced by GCA (Fig. 7h). Nuclear translocation of NF-κB p65 has previously been shown to regulate the expression of various genes involved in the immune response and inflammation, including MHCII related gene[30,31], *Il1*, *Il6*, *Tnfa* etc[32]. Consistent with previous research, QPCR analysis implied that the expression levels of MHCII-related genes (*Ciita*, *Cd74*, and *H2-eb1*) and inflammation-related genes (*Il1b*, *Il6*, *Tnfa* and *Ccl2*) were strengthened in GCA treated group. Whereas inhibition of PAK1 activity abolished the function of GCA (Fig. 7i–m and Supplementary Fig. 7a–d). These data show that GCA binds to PHB2 to activate the PAK1-NF-κB signaling pathway, thus provoking an inflammatory response of adipocyte via not only innate immune (production of pro-inflammation factors) but also adaptive immunity response (the increased expression of MHCII-related genes).

### GCA-neutralizing antibody improves adipose tissue inflammation and insulin sensitivity in obese mice

Monoclonal antibody is one of the major scientific breakthroughs and plays a therapeutic role in a variety of diseases. In our previous study, we have successfully generated anti-GCA monoclonal antibodies, which showed the highest efficiency in blocking the effect of GCA[12]. These data prompted us to explore the effect of such anti-GCA neutralizing antibody on metabolic inflammation. The DIO mice were treated with various doses of GCA neutralizing antibody (GCA-NAb) via tail intravenous (i.v.) injection twice a week for 2 months, and body weight was simultaneously recorded. These mice did not differ from vehicle-treated controls concerning body weight (Supplementary Fig. 8a).

Low concentration GCA-NAb has no detectable effect on expression levels of inflammatory cytokine gene in eWAT, liver and adipocyte (Fig. 8a–c). However, mice treated with medium and high concentration GCA-NAb displayed lower expression levels of inflammatory cytokine gene in eWAT, liver and adipocyte compared with vehicle-treated controls (Fig. 8a–c). Moreover, the effect of high concentration is more remarkable (Fig. 8a–c). The numbers of macrophages infiltration in the eWAT were also decreased in medium-dose and high-dose GCA-NAb treated mice (Fig. 8d). Consistently, administration of medium-dose and high-dose GCA-NAb also inhibited M1 macrophage polarization (Fig. 8d). With the attenuation of inflammation, medium dose and high dose GCA-NAb mice manifested a decline in pro-inflammatory Th1 cells and an elevation of Treg cells (Fig. 8e, f). In addition, medium-dose and high-dose GCA-NAb-treated DIO mice also exhibited improved glucose homeostasis and insulin tolerance (Fig. 8g–l).

To further validate the therapeutic effect of GCA-Nab, we turned to ob/ob mice, as this is a well-accepted model of obesity. We injected ob/ob mice with GCA-NAb (1 mg/kg) via the tail vein twice a week for 2 months. The treatment of GCA-Nab contributed to reduced expression levels of inflammatory cytokine gene in eWAT, liver and adipocyte

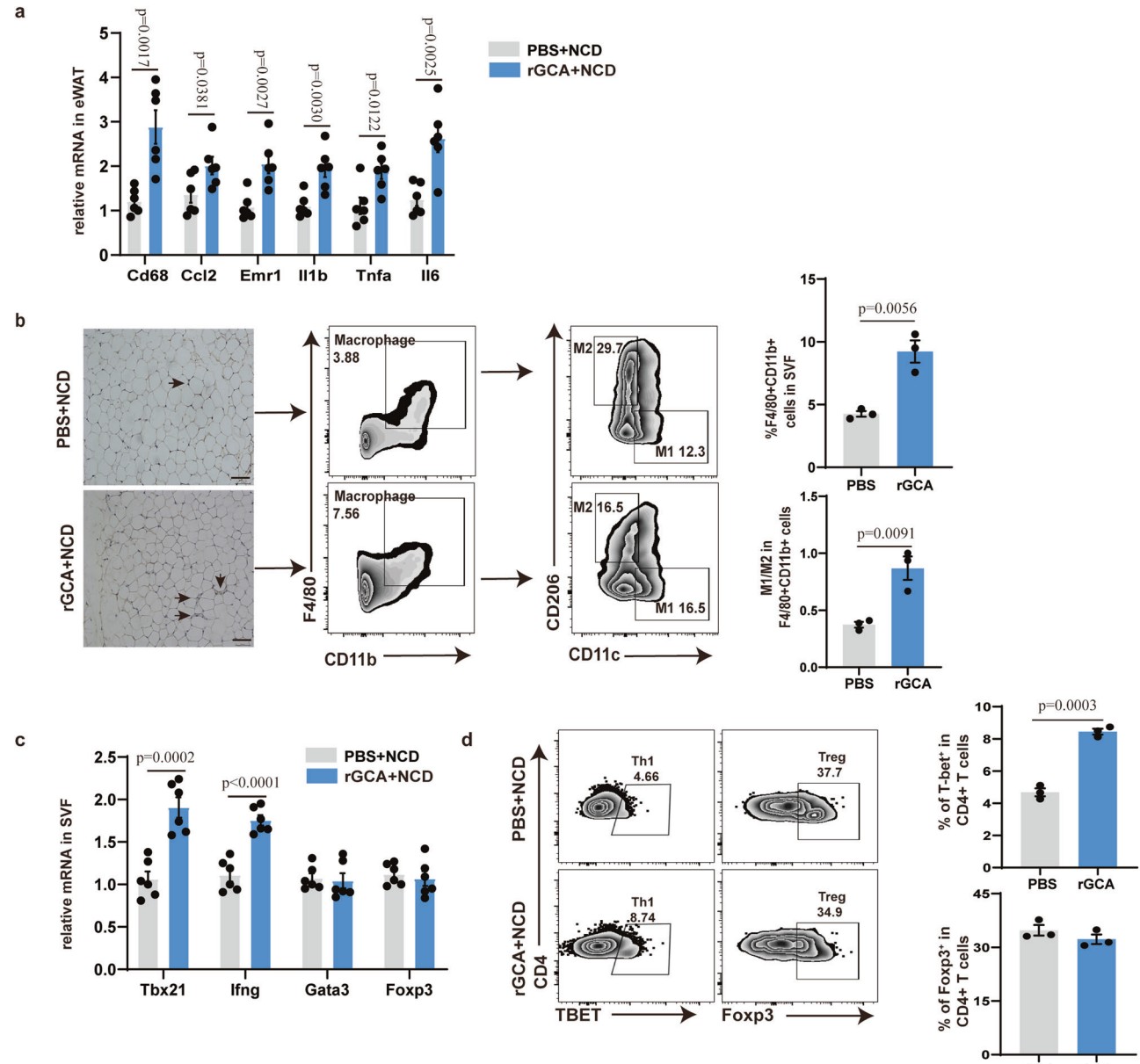

**Fig. 3 | GCA exacerbates adipose tissue inflammation in NCD-fed mice.**
**a** Inflammatory cytokine gene expression levels in eWAT from NCD-fed WT mice treated with PBS or rGCA ($n = 6$). **b** F4/80 immunohistochemistry and flow analyses of macrophage in eWAT from NCD-fed WT mice treated with PBS or rGCA ($n = 3$). Scale bar, 250 μm. **c** QPCR analysis of the proinflammatory Th1 marker genes (*Tbx21* and *Ifng*), Treg (*Foxp3*) and Th2 (Gata3) in SFV from NCD-fed WT mice treated with PBS or GCA ($n = 6$). **d** Flow cytometry analysis of CD4+Tbet+ Th1 cells and CD4+Foxp3+ Treg cells in eWAT from NCD-fed WT mice treated with PBS or GCA ($n = 3$). Data are presented as means ± SEM. n indicates the number of biologically independent samples examined. Statistical analysis was assessed by two-sided Student's $t$ test and significant differences were indicated with $p$ values. Source data are provided as a Source Data File.

(Supplementary Fig. 8b–d). Further, treatment with GCA-NAb reduced the infiltration of macrophages in eWAT and exhibited a lower ratio of M1 to M2(Supplementary Fig. 8e). Mice with supplementation of GCA-Nab also had decreased numbers of pro-inflammatory Th1 cells and a raise of Treg cells (Supplementary Fig. 8f, g). In line with improved adipose tissue inflammation, the insulin sensitivity was also ameliorated in the GCA-Nab treated group (Supplementary Fig. 8h–n). Altogether, these data suggest that GCA-neutralizing antibody improves adipose tissue inflammation and insulin sensitivity in obese mice.

## Discussion

Obesity-induced metabolic diseases involve functional integration among several organs via circulating factors, but little is known about the crosstalk between bone and adipose tissue. Herein, we report

myeloid-derived factor GCA to be an immunometabolic regulator that links obesity to inflammation.

Metabolic inflammation is a complex process. The exact agents that trigger obesity-related inflammation are still poorly understood, whereas several potential mechanisms have loomed. Signals derived from adipose tissue itself, such as adipokines[33] and cytokines[34], are capable of initiating the inflammatory response. Quite apart from that, factors originating from other organs, especially bone, also have a vital role in the regulation of metabolic homeostasis[35]. In our study, we found that myeloid-derived GCA is elevated in serum in both mice and human participants with obesity and increased within 2 weeks of HFD in mice. These indicated that GCA may have a crucial role in initiating adipose tissue inflammation. As expected, we found that genetic deletion of GCA in myeloid cells attenuated adipose tissue

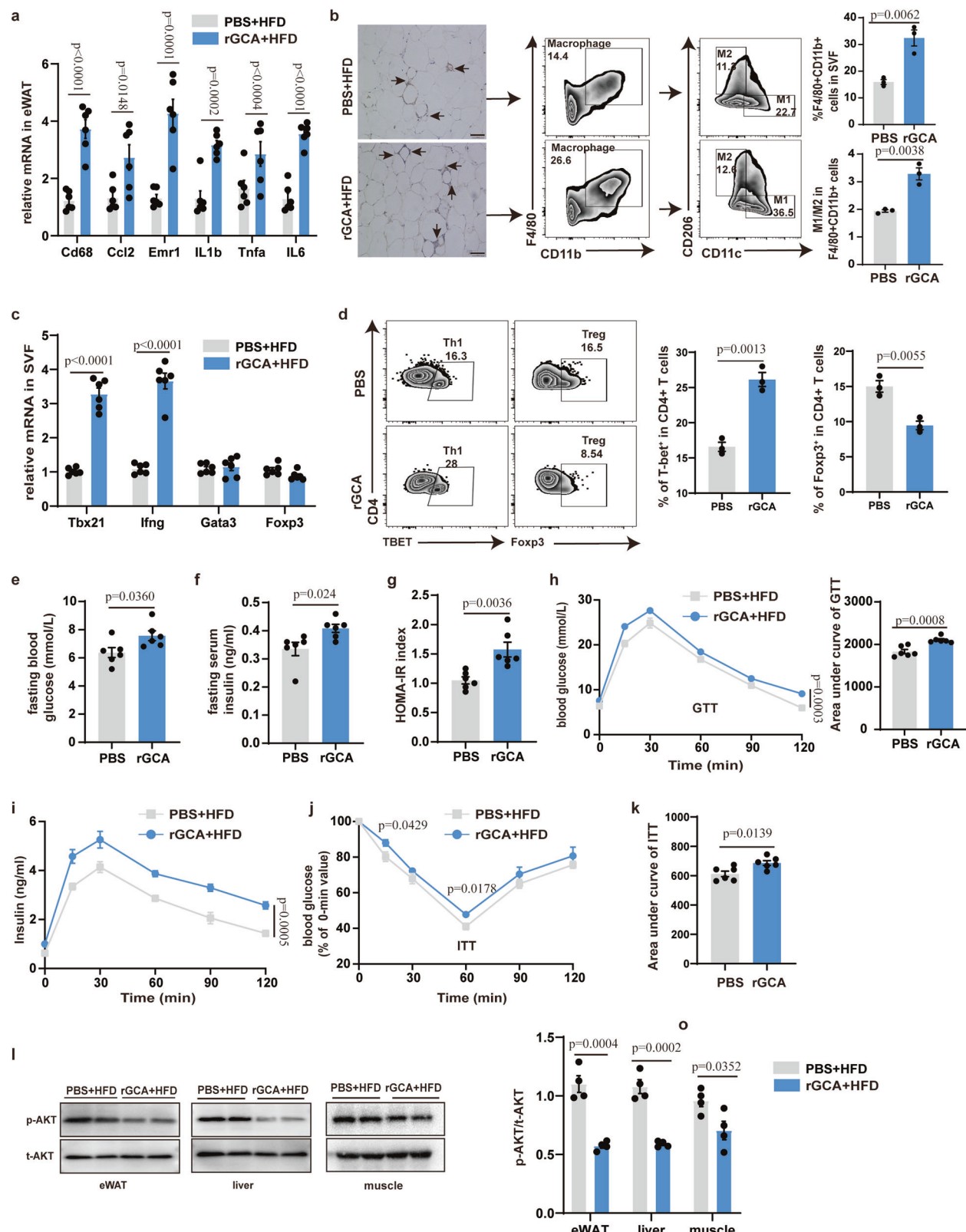

inflammation and metabolic dysfunction in obese mice, whereas injection of recombinant GCA into mice caused adipose tissue inflammation and insulin resistance. However, it is noteworthy that as there are many factors in circulation or local microenvironment regulating the function and metabolism of the whole body, GCA is among the many factors that could cause insulin resistance in obesity. Of note, our results also revealed that GCA expression levels display a positive

relationship with HOMA-IR, serum IL6, TNFα and MMP2 in human studies.

It is well recognized that adipocytes in adipose tissue are not merely a passive lipid-storing reservoir, but rather components of a dynamic endocrine organ primarily contributing to metabolic equilibrium and adipose tissue homeostasis[36,37]. Adipocytes dysfunction is an early pathophysiological change in the development of adipose

**Fig. 4 | GCA exacerbates adipose tissue inflammation, insulin resistance and glucose intolerance in HFD-fed mice. a** Inflammatory cytokine gene expression levels in eWAT from HFD-fed WT mice treated with PBS or rGCA (*n* = 6). **b** F4/80 immunohistochemistry and flow analyses of macrophage in eWAT from HFD-fed WT mice treated with PBS or rGCA (*n* = 3). Scale bar, 250 μm. **c** QPCR analysis of the proinflammatory Th1 marker genes (*Tbx21* and *Ifng*), Treg (*Foxp3*) and Th2(Gata3) in SFV from HFD-fed WT mice treated with PBS or GCA (*n* = 6). **d** Flow cytometry analysis of CD4+Tbet+ Th1 cells and CD4+Foxp3+ Treg cells in eWAT from HFD-fed WT mice treated with PBS or GCA (*n* = 3). **e–g** Fasting blood glucose, fasting insulin and HOMA-IR index of HFD-fed mice treated with PBS or rGCA (*n* = 6). **h–i** Time courses of blood glucose (**h**) and serum insulin concentrations (**i**) in HFD-fed mice treated with PBS or rGCA during an intraperitoneal glucose tolerance test (GTT)

(*n* = 6). **j** ITT of HFD-fed animals treated with PBS or rGCA (*n* = 6). The values for each line of the ITT graph represents the percentage of the measured concentration at time 0 for each group presented. **k** AUC of ITT (*n* = 6), in which the values of the y-axis are the absolute measured blood glucose concentrations. Western Blot of insulin-stimulated AKT phosphorylation in eWAT, liver and muscle and quantitation of pAKT/tAKT from HFD-fed animals treated with PBS or rGCA (*n* = 4). Data are presented as means ± SEM. n indicates the number of biologically independent samples examined. Statistical analysis was assessed by two-sided Student's *t* test (**a–g, h** (right), **k** and **l**) or two-way ANOVA followed with Sidak's multiple comparisons test (**h** (left)–**j**) and significant differences were indicated with *p* values. Source data are provided as a Source Data File.

tissue inflammation[38]. Therefore, we studied the effects of GCA on differentiated adipocytes in vitro, and found that GCA had a concentration- and time-dependent influence on proinflammatory-related gene expression in adipocytes. Moreover, multiple studies also suggest that adipocytes function as antigen-presenting cells in adipose tissue that activate pro-inflammatory Th1 cells[39]. Deng's research also indicated that induction of adipocyte MHCII contributes to the early activation and polarization of ARTs after the HFD challenge[22]. We further found that GCA upregulated MHCII expression in adipocytes via PHB2-PAK1-NF-κB signaling pathway, thus facilitating Th1 immunity response and IFN-γ production. Then, the activated pro-inflammatory T cell disrupt adipocyte homeostasis and provoke an immune response that results in further recruitment of circulating pro-inflammatory macrophages, therefore aggravating adipose tissue inflammation[20].

Although the effect of chronic inflammation on insulin resistance has been reasonably confirmed, the therapeutic avenues to improve this have not achieved enough success in clinical trials[40,41]. On other hand, the role of some inflammatory cytokines, such as IL6, in metabolic disease is both confusing and controversial. Although circulating IL-6 levels were increased in individuals with obesity and insulin-resistant[42], administration of recombinant IL-6 into humans improves insulin sensitivity[43]. Another research also found that IL6 deficiency aggravated hepatic insulin resistance and inflammation[44]. Based on our findings that GCA is a critical initiator of both innate and adaptive immune responses in adipose tissues early after overnutrition, one can raise the possibility that targeting GCA could be a future approach to the treatment of insulin resistance. As expected, we found that GCA-neutralizing antibody improves adipose tissue inflammation and insulin sensitivity in DIO mice. Our previous research also indicated that GCA-neutralizing antibody could ameliorate senile osteoporosis[12]. Pro-inflammatory cytokines can accelerate bone loss in menopausing women or the aging population, eventually leading to osteoporosis[45]. Together, these results provide a solid basis for the application of GCA in other inflammatory-related metabolic diseases.

In summary, we conclude that GCA is a myeloid-derived factor that can bind to the PHB2 receptor in adipocytes to activate PAK1- NF-κB signaling pathway, thus provoking adipose inflammatory response via not only innate immune (production of pro-inflammation factors) but also adaptive immunity response (the increased expression of MHCII) (Fig. 9). These observations define a mechanism whereby bone marrow factor GCA initiates adipose tissue inflammation and insulin resistance, showing that GCA could be a potential target to treat metainflammation.

## Methods
### Animal experiments
The study was approved by the Animal Care and Use Committees of the Laboratory Animal Research Center at Xiangya Medical School of Central South University. The animals received humane care according to the criteria outlined in the Guide for the Care and Use of Laboratory Animals prepared by the National Academy of Sciences and published

by the National Institutes of Health. *Gca*<sup>flox/flox</sup>(exons 3) mice were obtained from BIORAY LABORATORIES (China). *Lyz2*-Cre mice (Stock No: 004781-B6.129P2-*Lyz2*<sup>tm1(cre)Ifo</sup>/J) were purchased from Jackson Laboratory (USA). Male C57BL/6 J mice and ob/ob mice were obtained from Hunan SJA Laboratory Animal Company. *Gca*-floxed mice were bred with Lyz2-Cre mice to generate myeloid cell–specific KO mice and littermate (*Gca*<sup>+/+</sup>) control. All animals used in this research were kept in a specific pathogen-free (SPF) environment (12 h dark/light cycle (7:00-19:00 light on), constant and suitable room temperature (22-25 °C)) with free access to food and water. For HFD mice, male C57BL/6 J mice were fed with HFD (D12492, 60% kcal fat, 20% kcal carbohydrates, and 20% kcal protein; Wuhan BIOPIKE Bioscience Co. Ltd., Wuhan, China) or normal chow diet (NCD, sws9102, Jiangsu Xietong Pharmaceutical Bio-engineering Co., Ltd.) from the age of 8 weeks until the end of the experiment. Male C57BL/6 J were fed with HFD for 12 weeks to constructed DIO model. Only male mice were used for experiments. Mice were euthanized by carbon dioxide (CO2) asphyxiation inhalation and cervical dislocation was performed as a secondary euthanasia procedure, and then the tissues were isolated.

For the generation of adipocyte-specific *Phb2* knockdown mice, adeno-associated viral vector 9 (AAV9) which carries aP2 promoter-driven small hairpin RNAs targeting *Phb2* was injected intravenously with 5×10<sup>10</sup> genome copies into male C57BL/6 J mice aged 2-month-old. One month after the virus injection, the mice underwent metabolic phenotyping and were sacrificed for tissue collection and biochemical study.

### Glucose and insulin tolerance tests
The operation of GTTs and ITTs was performed as reported previously[46–48]. In brief, for GTTs, the mice were maintained an overnight (16 hours) fasting, and then the glucose (2 g/Kg body weight) was intraperitoneally injected into mice. For ITTs, the mice were kept 6 hours fasting, and then injected with insulin (0.75 units/Kg body weight) intraperitoneally. The blood glucose at different time points was monitored. The insulin was determined using an Ultra-Sensitive Insulin ELISA Kit (Crystal Chem, Downers Grove, IL). The HOMA-IR index was calculated as fasting blood glucose (mmol/L×fasting insulin(μU/L)/22.5.

### Isolation of mice primary stromal vascular fractions (SVF) and mature adipocytes
Isolation of mice primary stromal vascular fractions (SVF) and mature adipocytes was performed as reported previously[49]. eWAT (epidydimal white adipose tissue) was separated from male mice, and washed in PBS with 1% PS. Then put the chopped tissue in 2 mg/ml collagenase type 2 (C6885, Sigma-Aldrich) in digestion buffer (0.4 g/ l KCl, 0.06 g/l KH2PO4, 8 g/l NaCl, 0.09 g/l Na2HPO4•·7H2O, 1 g/l glucose, 1.2 mM CaCl2, 1 mM MgCl2, 0.8 mM ZnCl2, 3% BSA) for 30 min at 37 °C in a shaker (140 rpm). Adding an equal volume of DMEM (Procell Life Science&Technology Co.,Ltd) containing 10% fetal bovine serum (FBS) to stop the digestion of collagenase. The cell suspension was centrifuged at a speed of 300 g for 5 minutes. Mature adipocytes exist in

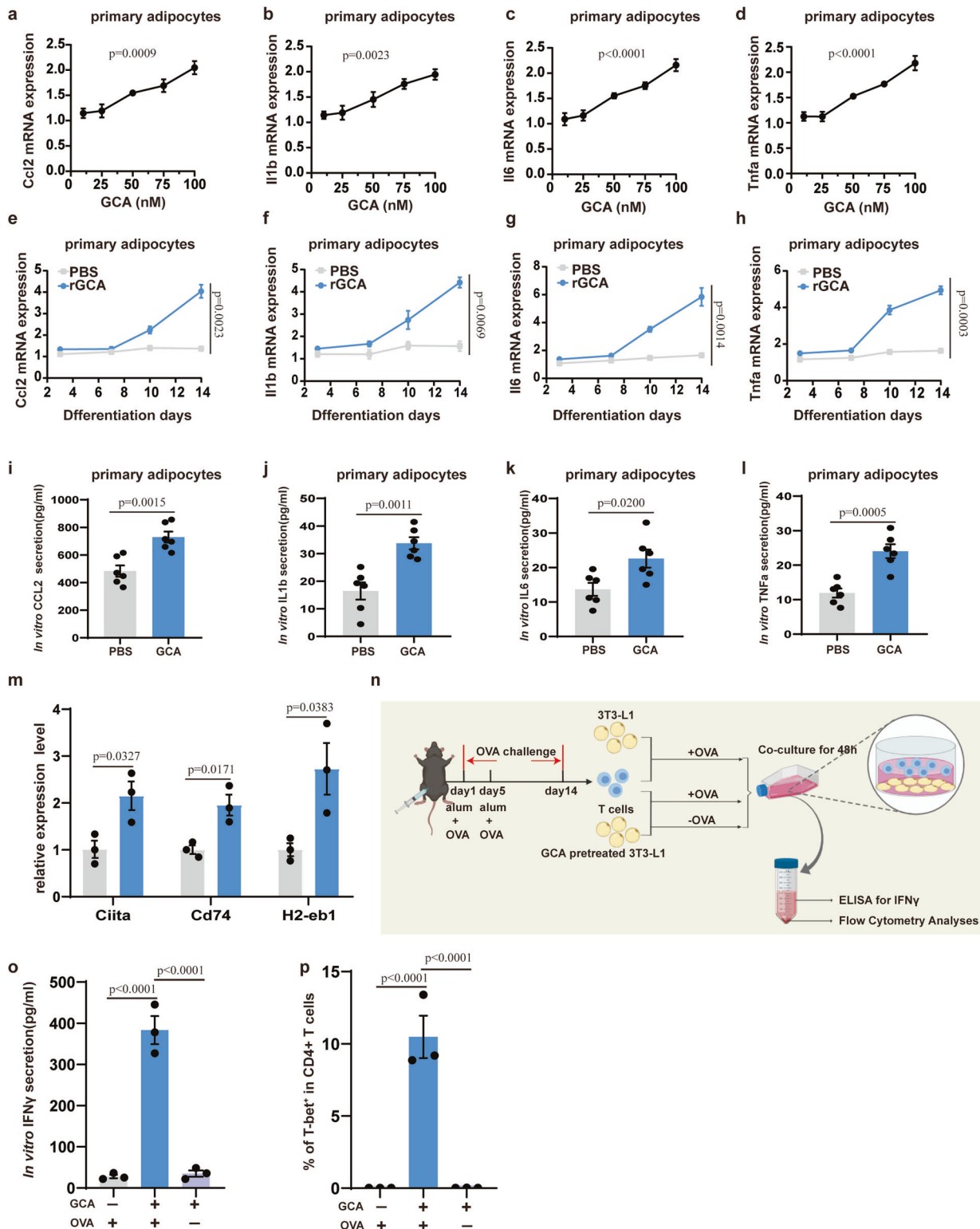

**Fig. 5 | GCA magnifies inflammation in adipocytes in vitro. a–d** QPCR analysis of *Ccl2* (**a**), *Il1b* (**b**), *Il6* (**c**) and *Tnfa* (**d**) in differentiated primary adipocytes administrated with GCA concentrations ranging from 10 to 100 nM (*n* = 3). **e–h** QPCR analysis of *Ccl2* (**e**), *Il1b* (**f**), *Il6* (**g**) and *Tnfa*(**h**) expression in cultured primary adipocytes administrated with 100 nM GCA or PBS analyzed at four time points during adipogenic induction differentiation process(*n* = 3). (**i–l**) The protein secretion of CCL2 (**i**), IL1b (**j**), IL6 (**k**) and TNFα (**l**) at day 14 of adipogenic induction differentiation process in primary adipocytes administrated with 100 nM GCA or PBS (*n* = 6). **m** QPCR analysis of MHCII family genes in primary adipocytes treated with GCA or PBS (*n* = 3). **n** A model illustrating 3T3-L1 adipocytes and T-cell co-

culture assay. This diagram was created with MedPeer.com. **o** ELISA of interferon γ (IFNγ) concentrations in the cell culture supernatants after co-culture assay (*n* = 3). **p** Flow cytometry analysis for the percentages of CD4+Tbet+ T cells after co-culture assay (*n* = 3). Data are presented as means ± SEM. *n* indicates the number of biologically independent samples examined. Statistical analysis was assessed by one-way ANOVA with Tukey's multiple-comparison test (**a–d**, **o** and **p**), two-sided Student's *t* test (**i–m**) or two-way ANOVA followed with Sidak's multiple comparisons test (**e–h**) and significant differences were indicated with *p* values. Source data are provided as a Source Data File.

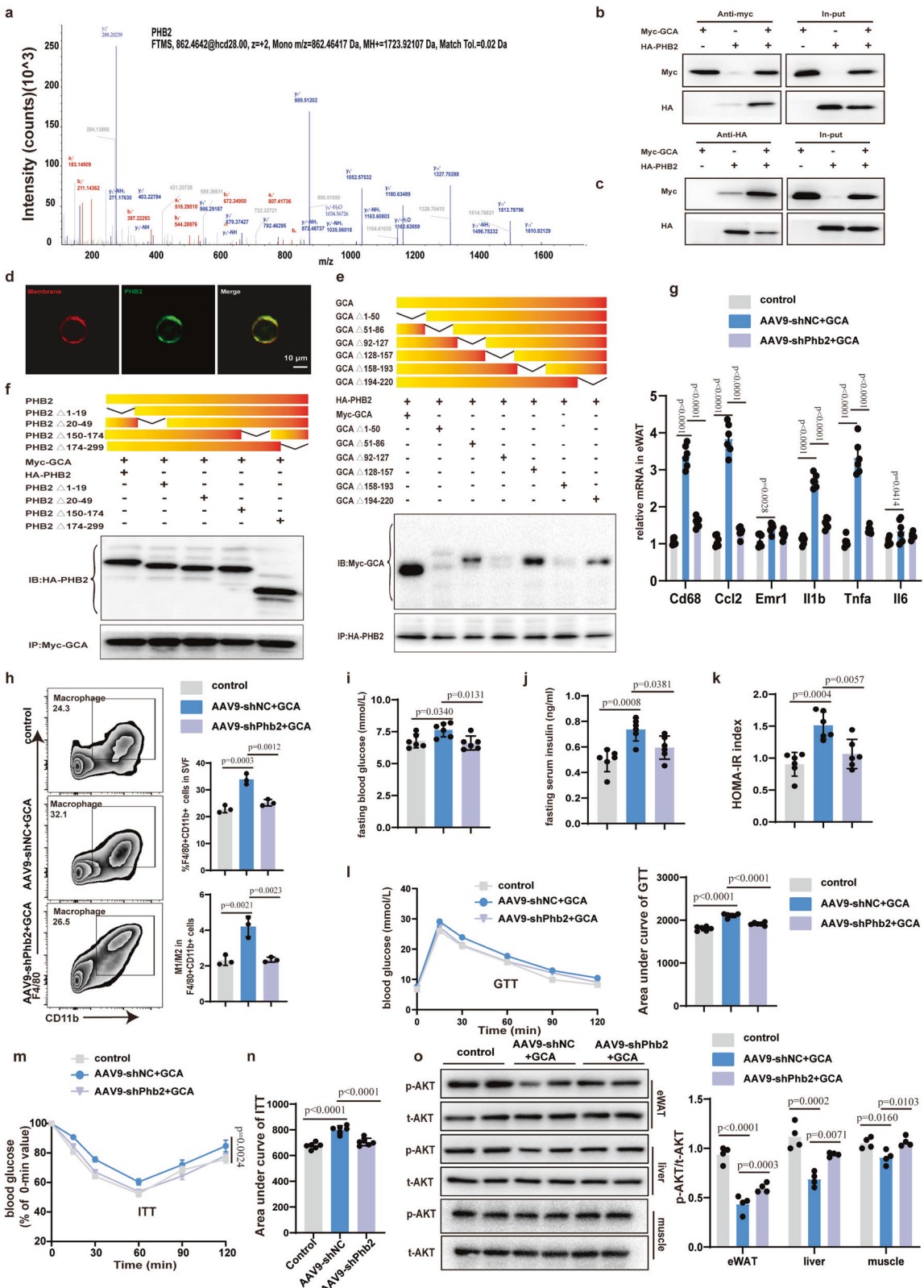

the supernatant, and SVF cells exist in the precipitates. Resuspending the pelleted cells and supernatant, respectively, and filtered through sterile 100 mm mesh filters. Then the SVF cells or mature adipocytes were plated on 6-well plates (NEST Biotechnology Co. Ltd (Wuxi, China)) in DMEM containing 10% FBS and 1% P/S at 37 °C in a humidified 5% CO2 incubator.

## The differentiation of 3T3-L1 preadipocytes

3T3-L1 preadipocytes were purchased from Procell Life Science & Technology Co. Ltd.(Wuhan,China). For the differentiation of 3T3-L1 preadipocytes, 48 hours after confluence, the 3T3-L1 cells were induced to differentiate with DMEM containing 10% FBS, 0.5 mmol/L IBMX (1-methyl-3-isobutylxanthine, I5879, Sigma-Aldrich), 1 μmol/L

**Fig. 6 | PHB2 is a functional receptor of GCA in adipocyte. a** The mass spectrum of prohibitin 2 (PHB2). **b–c** Immunoprecipitation (IP) analysis of Myc-GCA (**b**) and HA-PHB2 (**c**) binding. **d** Immunofluorescent staining of PHB2(green) in differentiated 3T3-L1. The cytoplasmic membrane was stained with Dil (red). Scale bars, 10 μm. **e** IP analysis of various GCA deletion mutants and their binding to full-length PHB2. **f** IP analysis of various PHB2 deletion mutants and their binding to full-length GCA. **g** QPCR analysis of inflammatory cytokine gene expression levels in eWAT of control mice and Phb2 knockdown mice with or without GCA treatment ($n = 6$). **h** Flow analyses of macrophage in eWAT from control mice, AAV9-shNC mice and AAV9-shPhb2 mice treated with GCA or PBS for 8 weeks ($n = 3$). **i–k** Fasting blood glucose, fasting insulin and HOMA-IR index of control mice, AAV9-shNC mice and AAV9-shPhb2 mice treated with GCA or PBS for 8 weeks ($n = 6$). **l, m** GTT and ITT of control mice, AAV9-shNC mice and AAV9-shPhb2 mice treated with GCA or PBS for 8 weeks ($n = 6$). (**n**) AUC of ITT ($n = 6$), in which the values of the y-axis are the absolute measured blood glucose concentrations. **o** Western Blot of insulin-stimulated AKT phosphorylation in eWAT, liver and muscle(left) and quantitation of pAKT/tAKT(right) from control mice, AAV9-shNC mice and AAV9-shPhb2 mice treated with GCA or PBS for 8 weeks ($n = 4$). Data are presented as means ± SEM. n indicates the number of biologically independent samples examined. Data shown in **b, c, d, e, f** are representative images of three independent experiments with similar results. Statistical analysis was assessed by one-way ANOVA with Tukey's multiple-comparison test (**g–k, l** (right), **n, o** (right) or two-way ANOVA followed with Sidak's multiple comparisons test (**m**) and significant differences were indicated with p values. Source data are provided as a Source Data File.

dexamethasone (D4902, Sigma-Aldrich), 10 μg/ml insulin and 2 μmol/L rosiglitazone for 3 days. Then the 3T3-L1 cells were cultured with DMEM supplemented with 10% FBS, 10 μg/ml insulin and 2 μmol/L rosiglitazone for another 2 days. After successful differentiation, follow-up experiments were carried out.

### 3T3-L1 and T-cell co-culture
C57BL/6 J mice were injected with 100 ug OVA and 1 mg Imject™ Alum Adjuvant at day 1 to induce OVA-specific T-cells. And then injecting with alum-OVA again at day 5. The mice were sacrificed at day 14. T cells in lymph nodes and spleen were isolated utilizing CD4$^+$ T Cell isolation Kits (Miltenyi Biotec). The GCA pretreated or untreated 3T3-L1 cells were cocultured with CD4$^+$ T Cells for 48 h in the presence/absence of 200 μg/ml OVA. Then, the cell supernatants were analyzed by ELISA and T cells were collected for flow cytometry analyses.

### Plasmid and siRNA transfection
The mouse *Gca* pcDNA3.1-MYC-C plasmid, mouse *Phb2* pcDNA3.1-HA-C plasmid, mouse *Phb2* pcDNA3.1-HA-N plasmid, mouse *Gca* delete 1-50 pcDNA3.1-MYC-C plasmid, mouse *Gca* delete 51-86 pcDNA3.1-MYC-C plasmid, mouse *Gca* delete 92-127 pcDNA3.1-MYC-C plasmid, mouse *Gca* delete 128-157 pcDNA3.1-MYC-C plasmid, mouse *Gca* delete 158-193 pcDNA3.1-MYC-C plasmid, mouse *Gca* delete 194-220 pcDNA3.1-MYC-C plasmid, mouse *Phb2* delete 1-19 pcDNA3.1-HA-C plasmid, and mouse *Phb2* delete 19-49 pcDNA3.1-HA-C plasmid, mouse *Phb2* delete 150-174 pcDNA3.1-HA-C plasmid, mouse *Phb2* delete 174-299 pcDNA3.1-HA-C plasmid were constructed by YouBio Technology Corporation (Shanghai, China). HEK 293 T was purchased from Procell Life Science & Technology Co. Ltd. (Wuhan,China).

For the transfection of siRNA or plasmid, cells were seeded in 6-well plates and transfected with lipofectamine 2000 (Thermo Scientific). Transfection efficiency or functional validation was detected by Western blot (WB) and quantitative RT-PCR (qPCR).

### Cellular membrane protein extraction
The cellular membrane protein extraction of 3T3-L1 was performed using a kit from BioVision Incorporated (USA) according to the manufacturer's instructions.

### ELISA
The concentrations of IL-1b, IL-6, CCL2, TNF-α and INF-γ in the medium/serum were measured using ELISA kits from Thermo Scientific or MULTI SCIENCE according to the manufacturer's instructions.

### Immunoprecipitation and Western blot analysis
Immunoprecipitation was performed as previously described[50]. In brief, the total cell lysates were collected and incubated with corresponding antibodies (HA, H9658, Sigma,1:200; MYC, 2276 S, Cell Signaling Technology,1:250) and protein A/G beads at 4 °C overnight. Immunoprecipitants were separated by SDS−PAGE and visualized by ECL Plus. For western blot, tissue or cell lysates were separated by SDS-PAGE and blotted on PVDF (polyvinylidene difluoride) membranes

(Millipore). Then the membranes were incubated with corresponding primary antibody (GCA, PA5-77127, Invitrogen, 1:1000; p-AKT, CST9271s, Cell Signaling Technology,1:1000; AKT, CST9272s, Cell Signaling Technology,1:1000; PAK1, 2602 T, Cell Signaling Technology,1:1000; Phospho-PAK1, 2601 T, Cell Signaling Technology,1:1000; Phospho-NF-κB p65, CST3033S, Cell Signaling Technology,1:1000; NF-κB p65, CST8242S, Cell Signaling Technology, 1:1000; PHB2, sc-133094, Santa Cruz,1:1000) at 4 °C overnight. And specific proteins were visualized by ECL Plus.

### Immunofluorescence staining
Immunofluorescence staining was conducted according to standard procedures. Paraffin sections of tissue or cell climbing sheets were incubated with specific primary antibodies (GCA, PA5-77127, Invitrogen,1:200; F4/80, ab6640, abcam,1:200; PHB2, sc-133094, Santa Cruz,1:200; P65, CST8242S, Cell Signaling Technology,1:400, Ly6g/6c,Biolegend,108403,1:200;Perilipin-1,Cell Signaling Technology,9349 S,1:200) at 4 °C overnight. After washing with PBS for 3 times, the sections or sheets were incubated with corresponding fluorescent secondary antibodies. The cell nuclei were labeled with DAPI. The cytoplasmic membrane was stained with Dil. The results were imaged by fluorescence microscope or confocal microscopy.

### Real-time PCR
For qRT-PCR analysis, the total RNAs from cultured cells or tissue were isolated by RNAex Pro RNA reagent (Accurate Biotechnology (Hunan) Co., Ltd) and performed on the Applied Biosystems QuantStudio3. Primer sequences are listed in Supplementary Table 2.

### Mass spectrometry analysis
Mass spectrometry analysis was conducted by PTM Bio, China. For in-gel tryptic digestion, gel pieces were destained in 50 mM NH4HCO3 in 50% acetonitrile (v/v) until clear. Gel pieces were dehydrated with 100 μl of 100% acetonitrile for 5 min, the liquid removed, and the gel pieces rehydrated in 10 mM dithiothreitol and incubated at 56 °C for 60 min. Gel pieces were again dehydrated in 100% acetonitrile, liquid was removed and gel pieces were rehydrated with 55 mM iodoacetamide. Samples were incubated at room temperature, in the dark for 45 min. Gel pieces were washed with 50 mM NH4HCO3 and dehydrated with 100% acetonitrile. Gel pieces were rehydrated with 10 ng/μl trypsin resuspended in 50 mM NH4HCO3 on ice for 1 h. Excess liquid was removed and gel pieces were digested with trypsin at 37 °C overnight. Peptides were extracted with 50% acetonitrile/5% formic acid, followed by 100% acetonitrile. Peptides were dried to completion and resuspended in 2% acetonitrile/0.1% formic acid. The tryptic peptides were dissolved in 0.1% formic acid (solvent A), directly loaded onto a home-made reversed-phase analytical column (15-cm length, 75 μm i.d.). The gradient was comprised of an increase from 6% to 23% solvent B (0.1% formic acid in 98% acetonitrile) over 16 min, 23% to 35% in 8 min and climbing to 80% in 3 min then holding at 80% for the last 3 min, all at a constant flow rate of 400 nl/min on an EASY-nLC 1000 UPLC system. The peptides were subjected to NSI source followed by

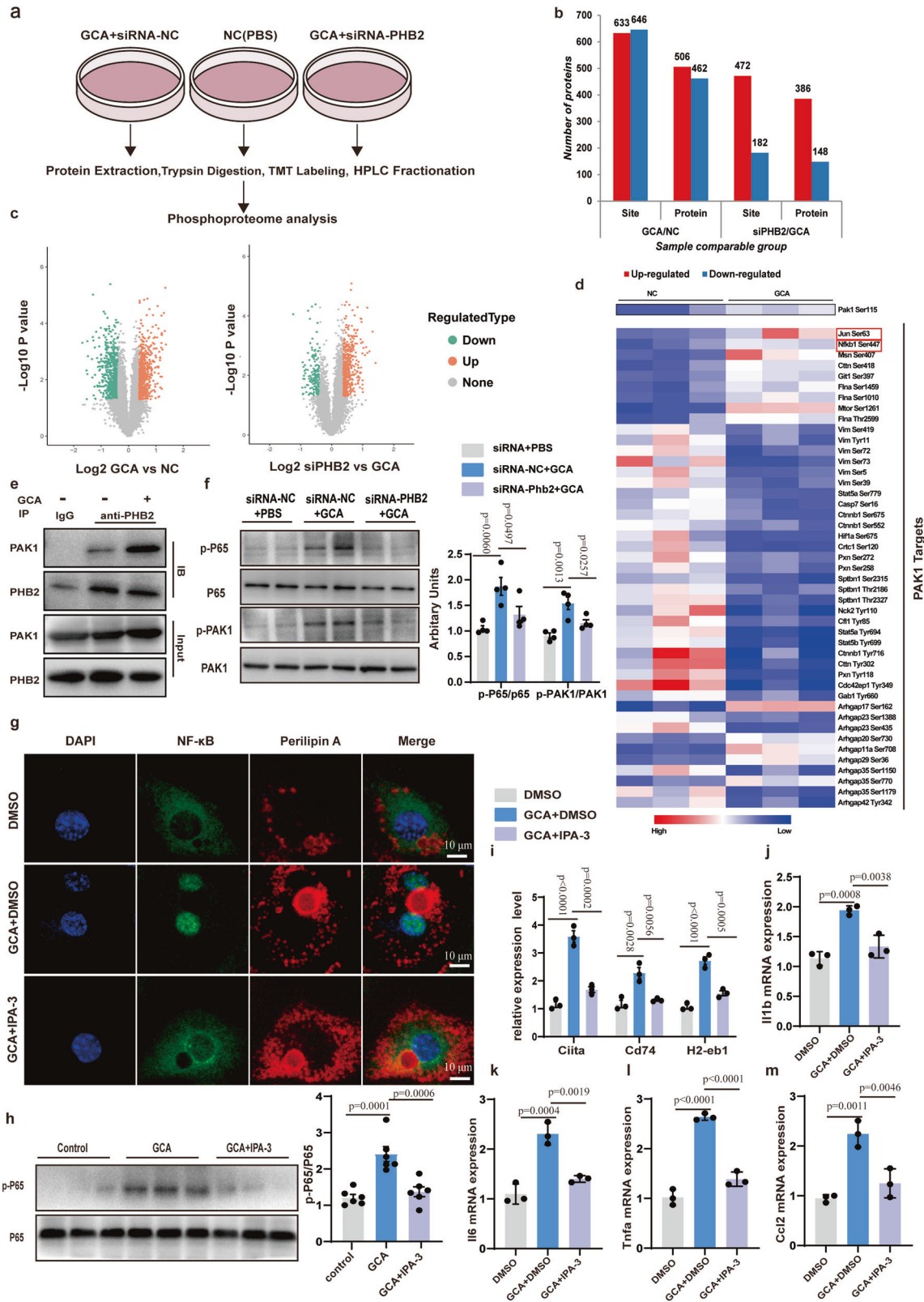

tandem mass spectrometry (MS/MS) in Q ExactiveTM Plus (Thermo) coupled online to the UPLC. The electrospray voltage applied was 2.2 kV. The m/z scan range was 350 to 1800 for full scan, and intact peptides were detected in the Orbitrap at a resolution of 70,000. Peptides were then selected for MS/MS using NCE setting as 28 and the fragments were detected in the Orbitrap at a resolution of 17,500. A data-dependent procedure that alternated between one MS scan followed by 20 MS/MS scans with 15.0 s dynamic exclusion. Automatic gain control (AGC) was set at 5E4. The resulting MS/MS data were processed using Proteome Discoverer 2.1. Trypsin/P (or other enzymes if any) was specified as cleavage enzyme allowing up to 2 missing cleavages. Mass error was set to 10 ppm for precursor ions and 0.02 Da for fragment ions. Carbamidomethyl on Cys were specified as fixed modification and oxidation on Metwas specified as variable

**Fig. 7 | GCA promotes phosphorylation of PAK1- NF-κB downstream of PHB2 signaling. a** Global quantitative phosphoproteomic analysis using differentiated 3T3-L1 adipocytes treated with GCA, PBS, or siRNA-PHB2. **b–c** Statistics (**b**) and volcano plot (**c**) of dysregulated phosphorylations. **d** Heatmap of the p21-activating kinase (PAK1) and its targets phosphorylation levels in the indicated groups. **e** Images of immunoprecipitation (IP) analysis using antibody target PHB2 followed by western blotting analysis using antibodies target PAK1 and PHB2. Data shown are representative images of three independent experiments with similar results. **f** Representative images of western blotting analysis of proteins/phosphorylations as indicated in differentiated 3T3-L1 adipocytes treated with GCA or siRNA-PHB2 (*n* = 3). **g** Representative immunofluorescent images of P65 (green) and Perilipin A(red) in differentiated 3T3-L1 adipocytes with addition of 10 μM IPA-3(PAK1 inhibitors) or DMSO and with or without GCA treatment (*n* = 3). The nuclei

were stained with DAPI. (*n* = 4). Scale bar, 10 μm. **h** Representative images of western blotting analysis of p-P65(left) and quantitation of p-P65/P65(right) in differentiated 3T3-L1 adipocytes with addition of 10 μM IPA-3(PAK1 inhibitors) or DMSO and with or without GCA treatment (*n* = 6). **i** QPCR analysis of MHCII family genes in differentiated 3T3-L1 adipocytes with addition of IPA-3 or DMSO and with or without GCA treatment (*n* = 3). **j–m** QPCR analysis of *Il1b*(**j**), *Il6* (**k**), *Tnfa* (**l**) and *Ccl2* (**m**) in differentiated 3T3-L1 with addition of IPA-3 or DMSO and with or without GCA treatment (*n* = 3). Data are presented as means ± SEM. **n** indicates the number of biologically independent samples examined. Statistical analysis was assessed by two-sided Student's *t* test (**c**) or one-way ANOVA with Tukey's multiple-comparison test (**f** and **h**–**m**) and significant differences were indicated with *p* values. Source data are provided as a Source Data File.

modification. Peptide confidence was set at high, and peptide ion score was set > 20.

## Phosphoproteome analysis

Phosphoproteome analysis was conducted by PTM Bio, China. Sample was sonicated three times on ice using a high intensity ultrasonic processor (Scientz) in lysis buffer (8 M urea, 1% protease inhibitor cocktail). The remaining debris was removed by centrifugation at 12,000 g at 4 °C for 10 min. Finally, the supernatant was collected and the protein concentration was determined with BCA kit according to the manufacturer's instructions. For digestion, the protein solution was reduced with 5 mM dithiothreitol for 30 min at 56 °C and alkylated with 11 mM iodoacetamide for 15 min at room temperature in darkness. The protein sample was then diluted by adding 100 mM TEAB to urea concentration less than 2 M. Finally, trypsin was added at 1:50 trypsin-to-protein mass ratio for the first digestion overnight and 1:100 trypsin-to-protein mass ratio for a second 4 h-digestion. Finally, the peptides were desalted by C18 SPE column. Tryptic peptides were firstly dissolved in 0.5 M TEAB. Each channel of peptide was labeled with their respective TMT reagent (based on manufacturer's protocol, ThermoFisher Scientific), and incubated for 2 hours at room temperature. Five microliters of each sample were pooled, desalted and analyzed by MS to check labeling efficiency. After labeling efficiency check, samples were quenched by adding 5% hydroxylamine. The pooled samples were then desalted with Strata X C18 SPE column (Phenomenex) and dried by vacuum centrifugation. The sample was fractionated into fractions by high pH reverse-phase HPLC using Agilent 300 Extend C18 column (5 μm particles, 4.6 mm ID, 250 mm length). Briefly, peptides were separated with a gradient of 2% to 60% acetonitrile in 10 mM ammonium bicarbonate pH 10 over 80 min into 80 fractions. Then, the peptides were combined into 9 fractions and dried by vacuum centrifuging. For the pan-antibody-based PTM enrichment, tryptic peptides dissolved in NETN buffer (100 mM NaCl, 1 mM EDTA, 50 mM Tris-HCl, 0.5% NP-40, pH 8.0) were incubated with pre-washed antibody beads at 4 °C overnight with gentle shaking. Then the beads were washed for four times with NETN buffer and twice with $H_2O$. The bound peptides were eluted from the beads with 0.1% trifluoroacetic acid. Finally, the eluted fractions were combined and vacuum-dried. For LC-MS/MS analysis, the resulting peptides were desalted with C18 ZipTips (Millipore) according to the manufacturer's instructions. For the bio-material-based PTM enrichment (for phosphorylation), Peptide mixtures were first incubated with IMAC microspheres suspension with vibration in loading buffer (50% acetonitrile/0.5% acetic acid). To remove the non-specifically adsorbed peptides, the IMAC microspheres were washed with 50% acetonitrile/0.5% acetic acid and 30% acetonitrile/0.1% trifluoroacetic acid, sequentially. To elute the enriched phosphopeptides, the elution buffer containing 10% $NH_4OH$ was added and the enriched phosphopeptides were eluted with vibration. The supernatant containing phosphopeptides was collected and lyophilized for LC-MS/MS analysis. The resulting MS/MS data were processed using MaxQuant search

engine (v.1.6.15.0). Tandem mass spectra were searched against the human SwissProt database (20422 entries) concatenated with reverse decoy database. Trypsin/P was specified as cleavage enzyme allowing up to 2 missing cleavages. The mass tolerance for precursor ions was set as 20 ppm in first search and 5 ppm in main search, and the mass tolerance for fragment ions was set as 0.02 Da. Carbamidomethyl on Cys was specified as fixed modification, and acetylation on protein N-terminal and oxidation on Met were specified as variable modifications. FDR was adjusted to <1%.

## Flow cytometry analyses

Mouse SVF or cell suspensions were incubated with BV510- Zombie Dyes for 15 mins at room temperature. After washing with PBS, CD45, CD4, CD8, F4/80, CD11c, and CD206 antibodies were added into the tube and incubated at 4 °C for 45 min. then the cells were washed with PBS again, and incubated with True-Nuclear™ 1X Fix Concentrate at 4 °C for 30 min. After washing with 1X Perm Buffer, T-bet and Foxp3 antibodies were added in the tube and incubated at 4 °C for 45 min. The cells were analyzed on BD FCACSCANTOII.

## Droplet-based scRNA-seq using the 10x genomics chromium platform

Bone marrow cells from NCD and HFD mice were flushed using ice-cold PBS, then dispelled into the single cell for further scRNA-Seq conducted by OE Biotech Co., Ltd (Shanghai, China). The scRNA-seq library was constructed using a 10X Genomics Chromium Single Cell 30 Reagent Kit v3 according to the user guide. The procedure samples a pool of ~3,500,000 10x Barcodes, and thousands of cells are partitioned into nanoliter-scale Gel Beads-in-emulsion (GEMs), where all generated cDNA shares a common 10x Barcode, thus separately index each cell's transcriptome. Libraries were sequenced and 10x Barcodes are used to associate individual reads back to the individual partitions. The 10X genomics raw data was processed using the Cell Ranger software pipeline (version 5.0.0) with default parameters and produced a matrix of gene counts versus cells. The R package Seurat (version 3.1.1) was used to process the unique molecular identifier (UMI) count matrix. Quality control was processed by filtering out these cells: 1) gene numbers less than 200, UMI less than 1000 and log10GenesPerUMI less than 0.7; 2) >10% of the counts belonged to mitochondrial genes and >5% of the counts belonged to hemoglobin genes. After filtering, we used functions from Seurat for downstream analysis. All gene expression was normalized and scaled using NormalizeData and ScaleData function. Top variable genes were identified by applying the FindVariableGenes function and using the PCA analysis. Cells were clustered using FindClusters function, and visualized using a 2-dimensional Uniform Manifold Approximation and Projection (UMAP) algorithm with the Run UMAP function. We used the FindAllMarkers function to identify marker genes of each cluster. Then, we used the R package SingleR to infer the cell of origin of each of the single cells independently and identify cell types.

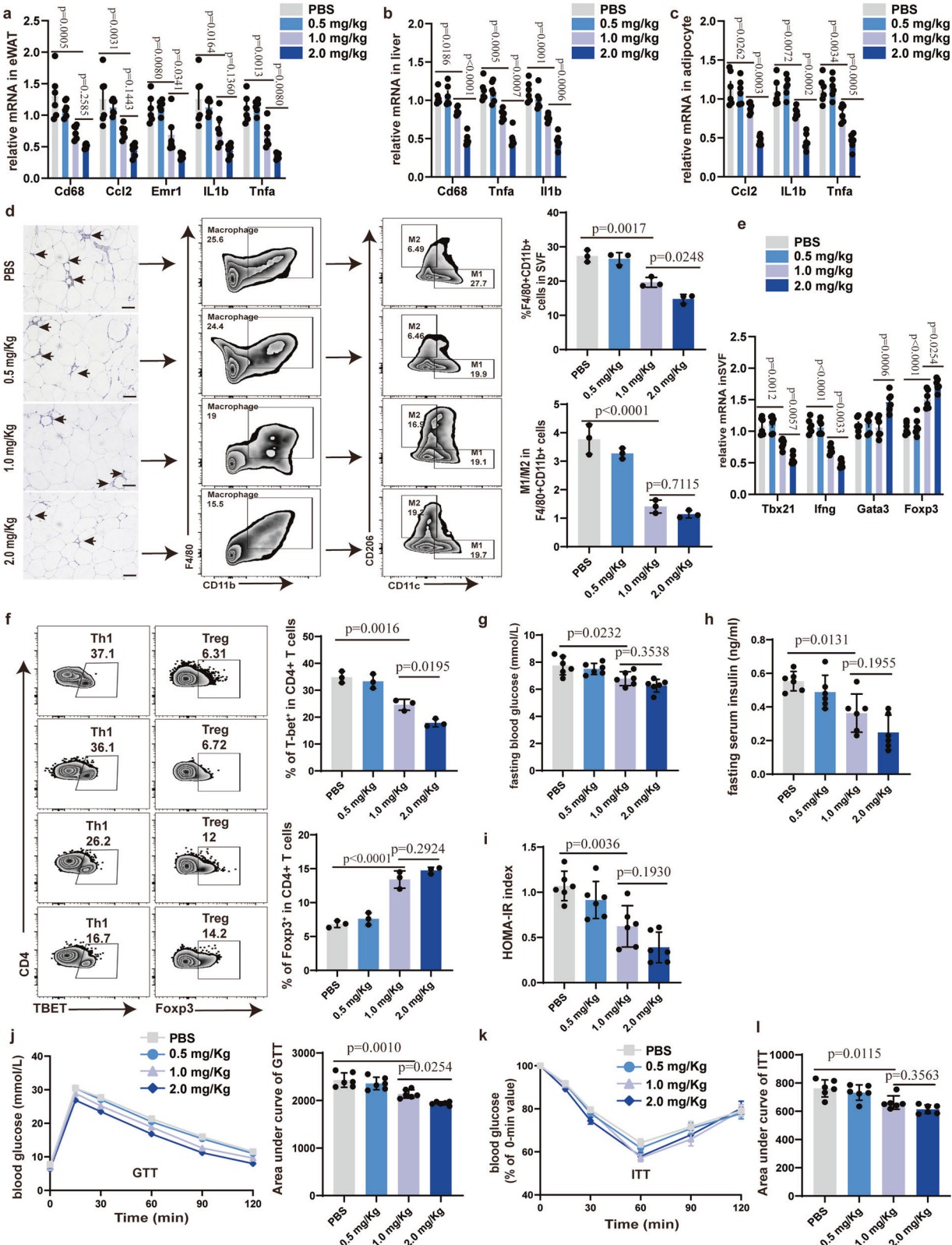

## GCA neutralizing antibody

The GCA neutralizing antibody was prepared by the Sino Biological.

## Clinical samples

All participants recruited in this study signed informed consent. The clinical studies were approved by the Ethics Committee of Xiangya Hospital at Central South University. The clinical characteristics of participants are summarized in Supplementary Table 1. The exclusion criteria for recruitment were thyroid diseases, autoimmune disease, taking medications, malignant tumors, other endocrinological diseases, such as Cushing syndrome, and vital organ failure, such as heart, liver or renal. No study participant received compensation.

**Fig. 8 | GCA-neutralizing antibody (GCA-NAb) improves adipose tissue inflammation and insulin sensitivity in obese mice. a–c** Inflammatory gene expression in eWAT(a), liver(b) and adipocyte(c) from DIO mice treated with PBS or different concentrations of GCA-NAb for 2 months (**n** = 6). **d** F4/80 immunohistochemistry and flow analyses of macrophage in eWAT from DIO mice treated with PBS or different concentrations of GCA-NAb for 2 months (*n* = 3). **e** Gene expression in SVF from diet-induced obesity (DIO) mice treated with PBS or different concentrations of GCA-NAb for 2 months (*n* = 6). **f** Flow cytometry analysis of CD4+Tbet+ Th1 cells and CD4+Foxp3+ Treg cells in eWAT from DIO mice treated with PBS or different concentrations of GCA-NAb for 2 months (*n* = 3). **g–i** Fasting blood glucose, fasting insulin and HOMA-IR index of DIO mice treated with PBS or different

concentrations of GCA-NAb for 2 months (*n* = 6). **j–k** GTT and ITT of DIO mice treated with PBS or different concentrations GCA-NAb for 2 months (*n* = 6). The values for the GTT graph are the absolute measured blood glucose concentrations, but for the ITT graph, the values for each different group represent the percentage of the initial measured blood glucose concentration at time 0 for that group. **l** AUC of ITT, in which the values of the y-axis are the absolute measured blood glucose concentrations (*n* = 6). Data are presented as means ± SEM. *n* indicates the number of biologically independent samples examined. Statistical analysis was assessed by one-way ANOVA with Tukey's multiple-comparison test and significant differences were indicated with *p* values. Source data are provided as a Source Data File.

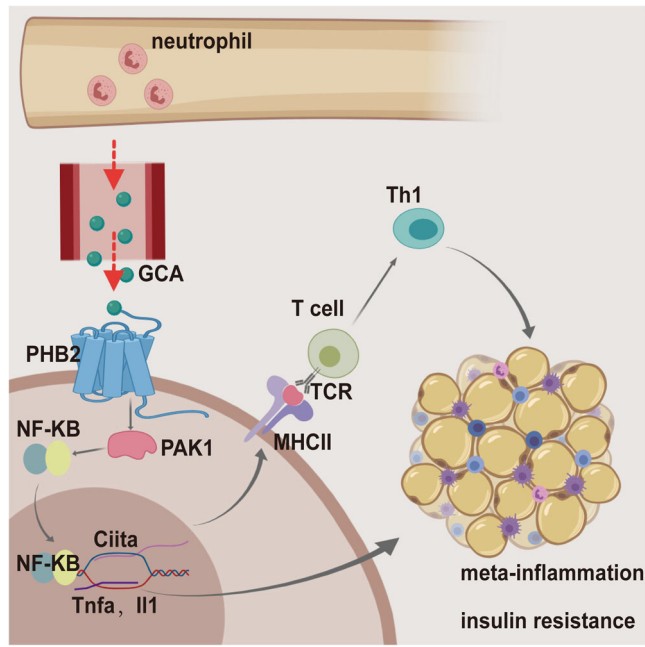

**Fig. 9 | Scheme of myeloid-derived grancalcin instigates obesity-induced insulin resistance and metabolic inflammation.** Obesity induces neutrophils and monocytes-macrophages producing copious amount of GCA. And GCA binds to the Prohibitin-2 (PHB2) receptor on adipocytes and activated the innate and adaptive immune response of adipocytes via PAK1-nuclear factor kappa-B (NF-κB) signaling pathway, thus provoking initiates adipose tissue inflammation and insulin resistance. This diagram was created with MedPeer.com.

## Statistical analysis

Data are presented as means ± SEM. All experiments were performed at least three times. The data are expressed as mean ± SEM. Two-tailed Student's *t* test was used to compare the two groups. When comparing the difference between multiple groups, one-way ANOVA with Tukey's multiple comparison test or two-way ANOVA were applied. Statistical differences were supposed to be significant when *P* < 0.05. The analysis was conducted using GraphPad 8.0 software. To analyze the correlation between GCA with HOMA-IR, IL6 TNFa and MMP2, two-sided Pearson's correlation test was applied.

Student's *t* test, one-way ANOVA, two-way ANOVA or Spearman's correlation was used to compare data between two or multiple groups when applicable. *P* < 0.05 was considered statistically significant. Data analyses were carried out by GraphPad Prism 8.0.

## Reporting summary

Further information on research design is available in the Nature Portfolio Reporting Summary linked to this article.

## Data availability

The scRNA-seq data from mouse samples produced in this paper have been deposited in the Sequence Read Archive database under accession number: SRR24235870, SRR24235869. Hyperlink: https://www.ncbi.nlm.nih.gov/sra/SRR24235870 https://www.ncbi.nlm.nih.gov/sra/SRR24235869 The mass spectrometry proteomics data have been deposited to the ProteomeXchange Consortium via the PRIDE partner repository with the dataset identifier PXD045836. Hyperlink: https://www.ebi.ac.uk/pride/archive/projects/PXD045836. Source data are provided in Source Data File with this paper. Source data are provided with this paper.

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

## Acknowledgements

This work was supported by the National Natural Science Foundation of China (grant nos. 81900732, 92149306, 82120108009, 82370883 and 81930022), the Natural Science Foundation of Hunan Province of China(2022JJ30957).

## Author contributions

X.-H.L., H.-Y. Z. and T.S. designed the experiments; T.S., Yue.H., Yan.H. and Y.X. carried out most of the experiments; Q.G., C.-J.L., L.-Y.C. and G.-P.C. helped to conduct animal experiments; M.-S.Y. helped to collect the clinical samples; X.-H.L. H.-Y. Z., T.S. and Yue.H. supervised the experiments, analyzed results, and wrote the manuscript.

## Competing interests

The authors declare no competing interests.
