## [Peer Review File · Nature Communications]

Myeloid-derived grancalcin instigates obesity-induced insulin resistance and metabolic inflammation in male miceREVIEWER COMMENTS

Reviewer #1 (Remarks to the Author):

Since this paper was transferred to Nature Communications, I have treated this as a revised manuscript review. The Authors have addressed many of the concerns and questions raised by the initial 3 Reviewers and have added a substantial amount of new data. As such, the manuscript is improved and I thought that the original version was also pretty good.

However, I still have a couple of residual questions.

1. The data on insulin sensitive vs. insulin resistant obese subjects is not that convincing. To my view, obesity is heterogeneous and insulin resistance spans a continuum across all obese patients from the least insulin resistant to the greatest. This is as expected and does not justify classification into a new category of ISOb. As I look at the data they presented, the ISOb patients are more insulin resistant than lean patients but less so than the IROb patients. In other words, this could just be an aspect of the spectrum of insulin resistance across obesity, and it would not be justified to carve out the low end of the spectrum vs. the high end of the spectrum to create 2 different categories. I don't think this changes any of the substantive messages of this paper, but it does point out that the division into IS and IR obese is rather artificial.

2. Given the near complete KO of GCA as shown in Supplemental Figure 2A, it seems as if the serum GCA levels in the KO mice should be decreased far more than is shown in Figure 2A, if the Authors hypothesis of myeloid cell origination of GCA is correct. This should be explored and carefully explained.

3. Based on the data shown and despite the Authors comments, it still looks to me like the effects of the GCA KO on GTTs and ITTs range from minimal to small. Based on this, I agree that GCA is among the many factors that could cause insulin resistance in obesity, but is not a "main" factor, but actually a contributing factor, among others. I think this should be clearly spelled out and the Authors should not try to overstate the importance of GCA, since it is not necessary for the scientific value of this paper.

Reviewer #4 (Remarks to the Author):

The revised manuscript has effectively addressed the concerns raised by the reviewers, leading to satisfactory resolutions for most of the issues. Nevertheless, the current results do not provide complete support for the authors' conclusion. In order to enhance the manuscript and maximize its impact on the metainflammation research community, it is crucial to address the following key points:

1. As previously acknowledged by the authors, it is important to recognize that GCA can be produced by macrophages and neutrophils outside the bone marrow. Since obesity significantly upregulates neutrophil levels in adipose tissue, and the elevation of serum GCA coincides with the induction of neutrophils and macrophages in adipose tissue, it is imperative to investigate the potential role of adipose tissue neutrophils in serum GCA induction. Specifically, the authors should clarify whether there are any discernible differences in inflammatory function, such as inflammatory gene expression, between bone marrow neutrophils and adipose tissue neutrophils, or between GCA-positive and GCA-negative neutrophils in adipose tissue.
2. The authors previously reported that myeloid GCA primarily targets muscle in aging conditions. Considering the crucial role of muscle in regulating insulin sensitivity during obesity, it is important to examine the alterations in muscle function and morphology that may contribute to the GCA-mediated glucose regulation in obesity. Notably, in Fig 3o, 5m, and 7k, the ITT data suggests no significant differences among groups when the y-axis is expressed as the % baseline glucose. However, the insulin-induced phosphorylation of Akt was notably decreased in GCA-treated liver and adipose tissue. These discrepancies suggest that GCA may have differential effects on muscle insulin sensitivity, and this inconsistency should be addressed and reconciled.
3. The authors have effectively demonstrated the modulatory effects of GCA on adipose tissue and liver inflammation. To emphasize the tissue-specific effects of GCA, it would be beneficial to present data on the blood or lymph node leukocyte population.
4. The additional data presented in Figure 4, highlighting the adipocyte-specific function, is

commendable. However, to address concerns regarding potential myeloid cell contamination in the floating cell fractions, it is necessary to provide evidence of adipocyte and macrophage marker gene expression to confirm the specificity of the observed effects (Fig 2m, Fig 4n, Fig 7).

5. Although ob/ob and db/db mice are commonly used mouse models for studying obesity, it is important to note that their inflammatory profiles differ significantly. Therefore, considering that the authors employed ob/ob mice for therapeutic targeting (see Supplementary Figure 7), it would be more appropriate to assess the alteration of GCA levels in ob/ob mice, rather than db/db mice, in Figure 1.

6. In Figure 6g, the staining of perilipin A in the merged image does not correspond to the location of the surface layer of lipid droplets. This suggests that IPA-3, a small molecule PAK1 inhibitor, affects the lipid droplets in adipocytes, as evidenced by the absence of large lipid droplets in adipocytes, which is in stark contrast to the control or GCA+DMSO conditions. Therefore, it is necessary to verify whether this figure accurately represents the findings.

7. Considering that the increase in antigen presentation machinery genes in eWAT during obesity can result from both the rise of activated immune cells and adipocytes, it would be more appropriate to remove Figure 4m.

REVIEWER COMMENTS

Reviewer #1 (Remarks to the Author):

Since this paper was transferred to Nature Communications, I have treated this as a revised manuscript review. The Authors have addressed many of the concerns and questions raised by the initial 3 Reviewers and have added a substantial amount of new data. As such, the manuscript is improved and I thought that the original version was also pretty good. However, I still have a couple of residual questions.

1. The data on insulin sensitive vs. insulin resistant obese subjects is not that convincing. To my view, obesity is heterogeneous and insulin resistance spans a continuum across all obese patients from the least insulin resistant to the greatest. This is as expected and does not justify classification into a new category of ISOb. As I look at the data they presented, the ISOb patients are more insulin resistant than lean patients but less so than the IROb patients. In other words, this could just be an aspect of the spectrum of insulin resistance across obesity, and it would not be justified to carve out the low end of the spectrum vs. the high end of the spectrum to create 2 different categories. I don't think these changes any of the substantive messages of this paper, but it does point out that the division into IS and IR obese is rather artificial.

Response: In response to the reviewer's constructive suggestion, we are no longer divide obese patients into IS and IR in the revised manuscript. we compared the serum GCA between lean and obese subjects, and found that obese patients showed even higher serum GCA contents than lean subjects (Fig.1g and clinical data in Table S1).

2. Given the near complete KO of GCA as shown in Supplemental Figure 2A, it seems

as if the serum GCA levels in the KO mice should be decreased far more than is shown in Figure 2A, if the Authors hypothesis of myeloid cell origination of GCA is correct. This should be explored and carefully explained.

Response: This question is highly appreciated, which would improve our manuscript a lot. To explore the origination of GCA, we detected the expression level of GCA protein in various tissues from mice fed with NCD or HFD. The results showed that GCA protein levels were highly expressed in bone marrow, lowly expressed in adipose tissue, liver and muscle, but nearly non-existent in the spleen (Supplementary Fig. 1a-b). This was in line with previous studies^{1,2}. Moreover, the expression level of GCA was higher in bone marrow from HFD mice than NCD subjects. Compared with other tissues, bone marrow GCA rose most significantly after HFD challenge (Supplementary Fig. 1a-b). Precisely, myeloid cells are the main, rather than only source of circulating GCA in obesity.

- (1) Roes J, Choi BK, et al. Granulocyte function in grancalcin-deficient mice. *Mol Cell Biol.* 2003 Feb;23(3):826-30.
- (2) Liu F, Shinomiya H, et al. Characterization of murine grancalcin specifically expressed in leukocytes and its possible role in host defense against bacterial infection. *Biosci Biotechnol Biochem.* 2004 Apr;68(4):894-902.

3. Based on the data shown and despite the Authors comments, it still looks to me like the effects of the GCA KO on GTTs and ITTs range from minimal to small. Based on this, I agree that GCA is among the many factors that could cause insulin resistance in obesity, but is not a “main” factor, but actually a contributing factor, among others. I think this should be clearly spelled out and the Authors should not try to overstate the importance of GCA, since it is not necessary for the scientific value of this paper.

Response: Thanks for the reviewer's careful reading of our manuscript and offering valuable suggestions. We agree with the reviewer's idea that GCA is among the many factors that could cause insulin resistance in obesity, but is not a "main" factor, but actually a contributing factor, among others. Therefore, we have clearly spelled out this in the discussion in the revised manuscript.

Reviewer #4 (Remarks to the Author):

The revised manuscript has effectively addressed the concerns raised by the reviewers, leading to satisfactory resolutions for most of the issues. Nevertheless, the current results do not provide complete support for the authors' conclusion. In order to enhance the manuscript and maximize its impact on the meta-inflammation research community, it is crucial to address the following key points:

1. As previously acknowledged by the authors, it is important to recognize that GCA can be produced by macrophages and neutrophils outside the bone marrow. Since obesity significantly upregulates neutrophil levels in adipose tissue, and the elevation of serum GCA coincides with the induction of neutrophils and macrophages in adipose tissue, it is imperative to investigate the potential role of adipose tissue neutrophils in serum GCA induction. Specifically, the authors should clarify whether there are any discernible differences in inflammatory function, such as inflammatory gene expression, between bone marrow neutrophils and adipose tissue neutrophils, or between GCA-positive and GCA-negative neutrophils in adipose tissue.

Response: We appreciate very much for your professional review work on our manuscript. The reviewer is right that obesity significantly upregulates neutrophil levels in adipose tissue. However, the proportion of neutrophils in adipose tissue (which is account for 0.08% of SVF in NCD mice and 1.89% in HFD mice¹) is much lower than that in bone marrow (about 50-70%²) and circulation (10–25% of circulating leukocytes are neutrophils³). Moreover, bone marrow is a reservoir of mature neutrophils, which can be rapidly deployed to sites of inflammation or infection, such as adipose tissue⁴. Therefore, though adipose tissue infiltrating neutrophils may also contributes to serum GCA, its contribution is much negligible to some extent when compared with that from the bone marrow immune cells.

We made our efforts to isolate adipose neutrophils and compare their gene expression

profile with bone marrow neutrophils. However, as we mentioned above, the proportion of neutrophils in adipose tissue is quite low. Besides, neutrophils are short-lived and a large number of adipose neutrophils encountered apoptosis during the adipose disassociation and flow sorting process. Thus, we failed to clarify whether there are any discernible differences in inflammatory function between bone marrow neutrophils and adipose tissue neutrophils. Since adipose infiltrating neutrophils are derived from the bone marrow, we speculate that bone marrow neutrophils and adipose tissue neutrophils probably share similar transcriptome profiles.

On the other hand, we characterized the transcriptional differences in inflammatory function between bone marrow GCA-positive and GCA-negative neutrophils to represent GCA-positive and GCA-negative neutrophils in adipose tissue. The results showed that compared with GCA-negative neutrophils, GCA-positive neutrophils express higher levels of inflammation and immune-related genes (Supplementary Fig. 1h).

- (1) Talukdar S, Oh DY, et al. Neutrophils mediate insulin resistance in mice fed a high-fat diet through secreted elastase. *Nat Med.* 2012 Sep;18(9):1407-12.
- (2) Zhong J, Mao X, et al. Single-cell RNA sequencing analysis reveals the relationship of bone marrow and osteopenia in STZ-induced type 1 diabetic mice. *J Adv Res.* 2022 Nov;41:145-158.
- (3) Mestas J, Hughes CC. Of mice and not men: differences between mouse and human immunology. *J Immunol.* 2004 Mar 1;172(5):2731-8.
- (4) Kolaczowska E, Kubes P. Neutrophil recruitment and function in health and inflammation. *Nat Rev Immunol.* 2013 Mar;13(3):159-75.

2. The authors previously reported that myeloid GCA primarily targets muscle in aging conditions. Considering the crucial role of muscle in regulating insulin sensitivity during obesity, it is important to examine the alterations in muscle function and morphology that may contribute to the GCA-mediated glucose regulation in obesity. Notably, in Fig 3o, 5m, and 7k, the ITT data suggests no significant

differences among groups when the y-axis is expressed as the % baseline glucose. However, the insulin-induced phosphorylation of Akt was notably decreased in GCA-treated liver and adipose tissue. These discrepancies suggest that GCA may have differential effects on muscle insulin sensitivity, and this inconsistency should be addressed and reconciled.

Response: In fact, our previous research published on Cell Metabolism investigated the role of myeloid GCA on bone in aging conditions, rather than muscle. GCA repressed osteogenesis and promoted adipogenesis via acting on BMSCs¹. However, the reviewer is right that muscle also played a crucial role in regulating insulin sensitivity during obesity, it is important to examine the alterations in muscle function and morphology that may contribute to the GCA-mediated glucose regulation in obesity. Therefore, to examine the alterations in muscle function and morphology, we have performed HE staining on muscle and detected the insulin-induced phosphorylation of Akt in muscle. As shown Fig 2j, 2k and 3q, the insulin-induced phosphorylation of Akt was also increased in muscle from Gca-Lyz2-CKO mice and decreased in muscle from GCA treated mice, although the differences were not as pronounced as that in liver and adipose tissue. Whereas, deletion of Gca in myeloid cells or rGCA treatment displayed no effects on muscle morphology as revealed by HE staining (Supplementary Fig 2e and Supplementary Fig 3n).

According to the reviewer's suggestion, we have changed the y-axis of ITT data as the % baseline glucose. As shown in Fig 3o, and 7k, the blood glucose still had significant difference between two groups. To better illustrate the results of ITT, we still retained the format of presenting ITT results as AUC, in which the values of the y-axis are the absolute measured blood glucose concentrations. Consistently, the AUC of ITT also showed significant difference between two groups (Fig 3p) and between PBS group and 1.0 mg/Kg group (Fig.7l).

As for the Fig 5m, to assure the accuracy of the result, we have repeated the ITT assay.

As shown in the renewed data, insulin sensitivity was decreased in GCA treated mice, whereas the knockdown of Phb2 alleviated insulin intolerance in GCA treated mice. The insulin-induced phosphorylation of Akt in muscle was also consistent with the ITT results (Fig 5o). Altogether, these renewed figures with repeated data are much more evident indicating a deteriorated effect of GCA on insulin sensitivity.

(1) Li CJ, Xiao Y, et al. Senescent immune cells release grancalcin to promote skeletal aging. *Cell Metab.* 2021 Oct 5;33(10):1957-1973.

3. The authors have effectively demonstrated the modulatory effects of GCA on adipose tissue and liver inflammation. To emphasize the tissue-specific effects of GCA, it would be beneficial to present data on the blood or lymph node leukocyte population.

Response: Many thanks for your opinions. Indeed, GCA, as a secreted protein, exists in circulation. It can not only act on adipose tissue and liver, but also on other tissues, such as brain and spleen. Our manuscript was mainly focused on adipose tissue and liver inflammation, but our colleagues also investigated the role of GCA on brain or spleen. Therefore, we think that although presentation of data on the blood or lymph node leukocyte population will be helpful of illustration the effects of GCA on immune system, it is beyond our research. However, it is a thought-provoking idea that worthy of further investigation. On the other hand, previous research has indicated that the absence of GCA did not affect the generation of mature neutrophils in the bone marrow¹.

(1) Roes J, Choi BK, et al. Granulocyte function in grancalcin-deficient mice. *Mol Cell Biol.* 2003 Feb;23(3):826-30.

4. The additional data presented in Figure 4, highlighting the adipocyte-specific function, is commendable. However, to address concerns regarding potential

myeloid cell contamination in the floating cell fractions, it is necessary to provide evidence of adipocyte and macrophage marker gene expression to confirm the specificity of the observed effects (Fig 2m, Fig 4n, Fig 7).

Response: In response to the reviewer's constructive suggestion, we detected the expression level of adipocyte marker (Adipoq and Fabp4) and macrophage marker (Emr1 and Adgre1) via QPCR in the floating cell fractions. However, Emr1 and Adgre1 were below the detection limit in the floating cell fractions, which indicated that the proportion of macrophages is too low to detect. Moreover, the isolation of mice primary mature adipocytes was conducted as previously reported^{1,2}. The purity of the mature adipocytes obtained according to this method is relatively high.

Fig 1 QPCR analysis of adipocyte marker (Adipoq and Fabp4) and macrophage marker (Emr1 and Adgre1) in the floating cell fractions

- (1) Zhao GN, Tian ZW, et al. TMBIM1 is an inhibitor of adipogenesis and its depletion promotes adipocyte hyperplasia and improves obesity-related metabolic disease. *Cell Metab.* 2021 Aug 3;33(8):1640-1654.
- (2) Villanueva-Carmona T, Cedó L, et al. SUCNR1 signaling in adipocytes controls energy metabolism by modulating circadian clock and leptin expression. *Cell Metab.* 2023 Apr 4;35(4):601-619.

5. *Although ob/ob and db/db mice are commonly used mouse models for studying obesity, it is important to note that their inflammatory profiles differ significantly. Therefore, considering that the authors employed ob/ob mice for therapeutic targeting (see Supplementary Figure 7), it would be more appropriate to assess the alteration of GCA levels in ob/ob mice, rather than db/db mice, in Figure 1.*

Response: Thanks for the reviewer's constructive suggestion, we have assessed the alteration of GCA levels in ob/ob mice in revised manuscript (Supplementary Figure 1).

6. **In Figure 6g, the staining of perilipin A in the merged image does not correspond to the location of the surface layer of lipid droplets. This suggests that IPA-3, a small molecule PAK1 inhibitor, affects the lipid droplets in adipocytes, as evidenced by the absence of large lipid droplets in adipocytes, which is in stark contrast to the control or GCA+DMSO conditions. Therefore, it is necessary to verify whether this figure accurately represents the findings.**

Response: We apologize for this confusion. In fact, IPA-3 treated group also had large lipid droplets. It's just the picture we presented didn't have large lipid droplets. Therefore, we have replaced the picture in revised manuscript.

7. *Considering that the increase in antigen presentation machinery genes in eWAT during obesity can result from both the rise of activated immune cells and adipocytes, it would be more appropriate to remove Figure 4m.*

Response: We really appreciate this constructive suggestion. We have removed

Figure 4m in revised manuscript.

REVIEWERS' COMMENTS

Reviewer #1 (Remarks to the Author):

This paper is now acceptable.

Reviewer #4 (Remarks to the Author):

The authors have addressed all my concerns and those of the other reviewers very thoroughly and to my satisfaction and I recommend publication in Nature Communications